# RippleNet: Learning Causal Maritime Dynamics for Forecasting Warning-Induced Ripple Effects

## Abstract

Maritime transportation networks, including cargo vessels, tankers, and passenger ships, are critical to global trade but remain highly vulnerable to disruptions such as extreme weather or security alerts. These events often trigger ripple effects, with cascading impacts extending far beyond the initial warning zones. Traditional spatio-temporal forecasting methods struggle to capture these dynamics due to their reliance on correlations rather than causal reasoning, particularly in maritime contexts. To address this challenge, we propose RippleNet, a novel causal spatio-temporal framework that explicitly models causal dependencies to predict port-to-port flow disruptions under warning-induced ripple effects. RippleNet comprises three key components: (i) a neural deconfounding module that employs causal adjustment techniques to disentangle genuine causal effects from spurious correlations, addressing confounding factors that arise when warnings simultaneously affect multiple maritime operational aspects, (ii) a continuous-time ODE module that simulates the propagation of disruptions across vessel networks, and (iii) LLM-generated warning vectors that quantify the multidimensional operational impacts of various warning types. Experiments on maritime flow datasets from East Asia and Northwest Europe show that RippleNet significantly outperforms state-of-the-art baselines under warning scenarios, while offering interpretable causal insights into heterogeneous vessel flow behavior.

## 1 Introduction

Maritime transportation networks form the backbone of global trade and mobility, with cargo vessels carrying over 90% of international freight, tankers transporting essential energy resources, and passenger ships facilitating human mobility across oceans He et al. (2019); Zhou et al. (2020). However, these diverse vessel networks exhibit remarkable vulnerability to disruptions, where localized warnings can trigger cascading effects across the entire system, affecting all vessel types simultaneously. The September 2022 typhoon "Nanmadol" exemplified this phenomenon: while primarily affecting Japanese waters, the resulting warnings caused vessel traffic reductions of 70% across cargo, tanker, and passenger operations in Tokyo Bay, 72-hour delays for container ships in Shanghai, oil tanker rerouting through alternative channels, passenger ferry cancellations affecting thousands of travelers, and unexpectedly, 15% throughput drops in Singapore—located 3,000 kilometers from the typhoon's path—impacting all three vessel categories.

This "ripple effect" reveals a fundamental challenge in maritime operations: warning impacts propagate through complex causal mechanisms that extend far beyond geographical proximity El Mekkaoui et al. (2023); Hu et al. (2022). Figure 1 demonstrates how warning signals (orange-red regions) disrupt normal shipping patterns, with the Nagoya-Tokyo route experiencing a 70% flow reduction during the warning period. Understanding and predicting these cascading disruptions is critical for maintaining supply chain resilience Li et al. (2023); Zhang et al. (2019), yet remains a significant scientific challenge Su et al. (2020); Xu & Zhang (2022); Zhang & Li (2022).

Existing approaches to maritime traffic prediction face inherent limitations when confronting warning-induced disruptions. Traditional methods, including time series models Williams (1999); Zhang et al. (2011) and graph neural networks Wu et al. (2019; 2020); Veličković et al. (2017),

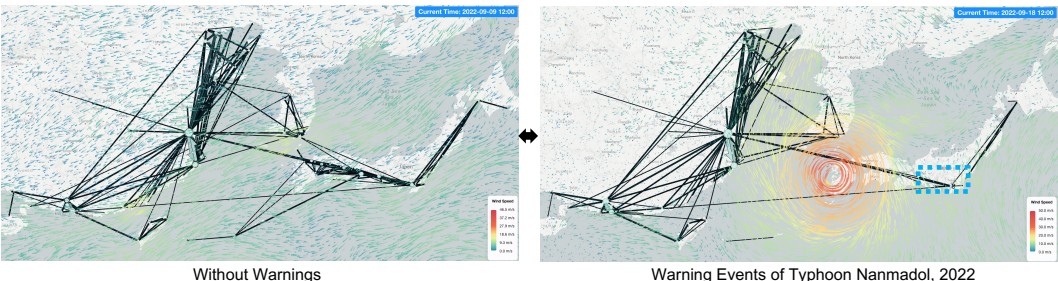

(a) Graphical illustration of the ripple effect in the maritime flow network. The left panel shows normal shipping patterns, while the right panel displays the network disruption during Typhoon Nanmadol, 2022. The orange-red heat map indicates warning intensity and demonstrates how warnings propagate through the network. The blue dashed box highlights the Nagoya-Tokyo shipping route.

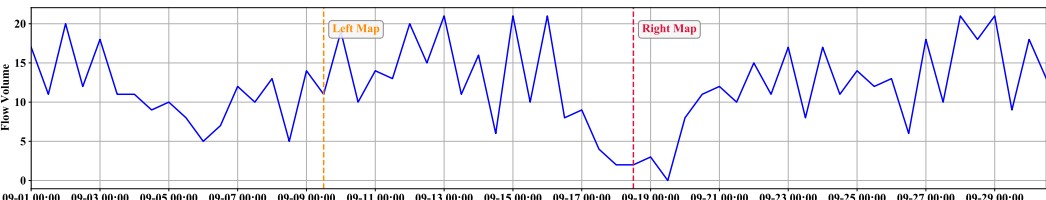

(b) Flow volume between the ports of Nagoya and Tokyo.

Figure 1: Motivation of our proposed framework.

rely primarily on historical correlations and perform adequately under normal conditions. However, they fail during anomalous events because they cannot distinguish between spurious correlations and genuine causal relationships Pearl et al. (2000); Glymour et al. (2016). Warning impacts exhibit pronounced spatio-temporal heterogeneity, where identical warnings produce vastly different effects across regions and time periods Zheng et al. (2020); Guo et al. (2019). This fundamental mismatch between correlation-based learning and causal propagation mechanisms results in substantial prediction degradation during critical warning scenarios Zhang & Li (2022); Zhou et al. (2022).

To address these limitations, we propose RippleNet (Causal Spatio-Temporal Transformer), a novel framework that explicitly models causal relationships in maritime warning propagation. Our approach represents inter-port dynamics as continuous-time systems governed by discovered causal structures Yang et al. (2022); Zheng et al. (2021); Fang et al. (2021); Ji et al. (2022). The main contributions of this work are summarized as follows:

- We propose a causally-informed deep learning framework for maritime flow prediction that integrates warning information through a neural deconfounder and continuous-time propagation dynamics.
- We develop an ODE-based ripple propagation module that captures both immediate and persistent disruption effects through adaptive decay mechanisms tailored to maritime operational constraints.
- We demonstrate substantial improvements over state-of-the-art baselines on two real-world maritime flow networks, with ablations verifying each component's contribution and case studies examining typhoon-driven ripple effects.

## 2 RELATED WORK

**Maritime Traffic Forecasting.** Maritime traffic prediction traditionally relies on statistical approaches like ARIMA Zhang et al. (2011), Kalman filtering He et al. (2019), and grey models Xiao & Duan (2020). Recent advances employ machine learning techniques, including PSO-BP networks Zhang et al. (2019) and deep learning Zhou et al. (2020); Li et al. (2023), showing improved accuracy. However, these methods primarily address normal operating conditions and fail to adequately model the impacts of maritime warnings or the cascading effects within port networks Su

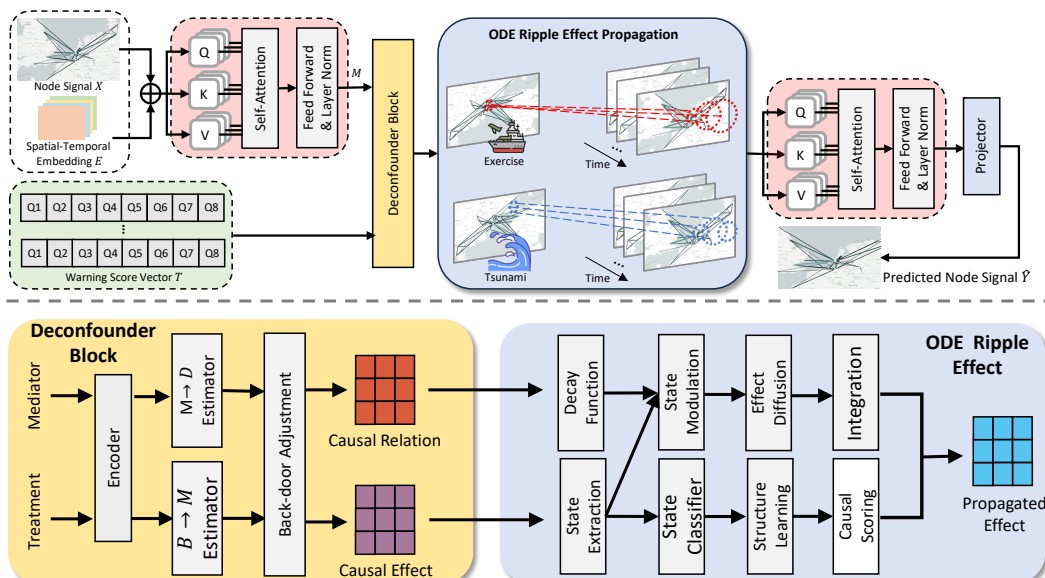

Figure 2: Architecture of our RippleNet framework. The model processes route signals and warning vectors through spatial-temporal embedding and attention mechanisms, decomposes causal relationships via learned deconfounder, and integrates ODE-based ripple effect propagation with decay functions to predict warning impacts across the maritime route network.

et al. (2020); Xu & Zhang (2022). Although some studies have attempted to incorporate weather data Hu et al. (2022), they do not effectively capture the complex causal mechanisms underlying ripple effects in maritime networks.

**Spatial-Temporal Graph Neural Networks.** STGNNs effectively model network dynamics Wu et al. (2019; 2020); Jin et al. (2023). Foundational works like STGCN Yu et al. (2018), Graph WaveNet Wu et al. (2019), and GMAN Zheng et al. (2020) capture spatio-temporal dependencies through graph convolutions, dilated convolutions, and attention mechanisms. Recent advances explore adaptive graph learning Bai et al. (2020), continuous-time modeling Fang et al. (2021); Choi et al. (2022), and multi-faceted spatial modeling Han et al. (2021). Despite success in road traffic forecasting Jiang et al. (2021), these correlation-based models struggle with anomalous events and warning scenarios Wang et al. (2021); Zhang et al. (2020).

**Causal Modeling for Complex Systems.** Correlation-based limitations have driven interest in causal inference for network data Pearl et al. (2000); Glymour et al. (2016). Recent integration of causal reasoning with deep learning includes causal attention Sui et al. (2022), deconfounding techniques Wu et al. (2022), and out-of-distribution frameworks Yang et al. (2022); Li et al. (2022); Zhou et al. (2022). While promising for robustness and interpretability, maritime applications remain unexplored. Our work introduces a causal-enhanced framework targeting ripple effect prediction following maritime warnings, incorporating higher-order topological structures Edelsbrunner et al. (2000); Huang et al. (2023) and causal discoveries to capture complex propagation patterns.

## 3 PRELIMINARIES

**Definition 1** (Maritime Network). *We represent the maritime network as a directed weighted time-varying graph $\mathcal{G}_t = (\mathcal{R}, \mathcal{E})$, where $\mathcal{R} = \{r_1, r_2, \ldots, r_N\}$ denotes the set of shipping routes, and $\mathcal{E} \subseteq \mathcal{R} \times \mathcal{R}$ represents the set of directed adjacency relationships between routes.*

**Definition 2** (Maritime Events). *Maritime events are defined as warnings or disruptions affecting shipping routes at specific times. Let $a_t^{r_i}$ denote an event affecting shipping route $r_i$ at time $t$. The system-wide event state is denoted by vector $A_t = [a_t^{r_1}, a_t^{r_2}, \ldots, a_t^{r_N}]^T$. We then obtain the*

**Maritime Warnings**

Weather   Drifting   Marine

Exercise   Navigation   Natural

**Structured Warning Information**
Location: Line Geometry
Time: 2022-03-01 00:00
Category: Exercise
Description: Potential ballistic missile launched from North Korea

**Prompted Engineering (0-100 Score)**

Q1: Spatial Impact?   Q2: Delay risk?
Q3: Reroute Need?   Q4: Duration impact?
Q5: Port Congestion?   Q6: Cargo threat?
Q7: Speed adjustment?   Q8: Uncertainty level?

Figure 3: **Prompt-based generation process of warning vector.** Maritime warnings are structured into location, time, category, and description, then processed by LLMs to generate quantitative impact scores (0-100) across eight dimensions.

*corresponding binary warning impact score $B_t$ via the Large Language Model (LLM), based on the input of maritime events $A_t$.*

**Definition 3** (Problem Formulation). *Given a maritime network represented as a time-varying graph $\mathcal{G}_t = (\mathcal{R}, \mathcal{E})$, where $\mathcal{R}$ denotes shipping routes and $\mathcal{E}$ represents directed route connections, our objective is to predict the ripple effects of maritime warnings on vessel flow patterns. We incorporate spatial-temporal embeddings $\mathbf{E}_t$ to capture node-specific contextual information. Formally, we learn a mapping:*

$$f : (\mathbf{X}_{t-T:t}, B_{t-T:t}, \mathbf{A}_{adj}, \mathbf{E}_{t-T:t}) \rightarrow \mathbf{Y}_{t+1:t+T'}$$

*where $\mathbf{X}_t \in \mathbb{R}^{N \times d_f}$ represents traffic flow features, $B_t \in \{0,1\}^8$ indicates binary warning impact scores, $\mathbf{A}_{adj} \in \{0,1\}^{N \times N}$ is the adjacency matrix encoding route connections, $\mathbf{E}_t \in \mathbb{R}^{N \times d_e}$ denotes spatial-temporal node embeddings, and $T, T'$ denote historical and prediction horizons respectively, with $N = |\mathcal{R}|$ being the number of shipping routes.*

## 4 METHODOLOGY

Building on the causal foundations established in our problem formulation, we present Rip-pleNet, a deep learning framework that operationalizes causal inference principles for maritime network prediction. As illustrated in Figure 2, our approach processes route signals and warning vectors through spatial-temporal embedding, applies a learned deconfounder for causal decomposition, and employs ODE-based ripple propagation to predict warning impacts across maritime networks. Throughout this section we explicitly treat the LLM-derived warning vector $B$ as the *operational treatment variable* and design the architecture to estimate $P(\mathbf{Y}_{t+1:t+T'} \mid \mathrm{do}(B))$ while adjusting for pre-treatment contextual covariates $(\mathbf{X}, \mathbf{V}, \mathbf{E})$. The

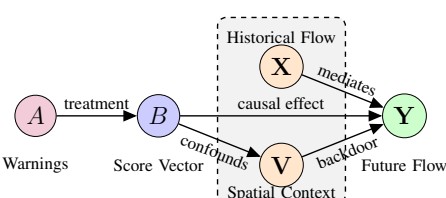

Figure 4: **Causal diagram illustrating the backdoor adjustment** paths from warning signal ($B$) to future flow ($\mathbf{Y}$) through key mediating mechanisms: historical flow patterns ($\mathbf{X}$) and adjacency relationships ($\mathbf{V}$), with maritime warning events ($A$) as the initial causal input.

framework addresses a critical limitation in existing methods: their dependence on correlations rather than causal mechanisms when modeling warning-induced disruptions Pearl et al. (2000); Yang et al. (2022).

**Warning Vector Generation.** Maritime warnings arrive in heterogeneous, unstructured formats—from meteorological bulletins describing typhoon trajectories to security advisories detailing piracy threats. This diversity creates challenges for neural networks requiring structured numerical inputs. We address this through large language model-based quantification using domain-specific prompts, as is shown in Figure 3.

Raw warning event $A_t$ is transformed into structured 8-dimensional impact scores $\mathbf{q}_t = [q_1^t, q_2^t, \ldots, q_8^t]^\top \in [0, 100]^8$, where each dimension captures distinct operational impacts: spatial coverage ($q_1$), delay risk ($q_2$), rerouting necessity ($q_3$), duration impact ($q_4$), route congestion ($q_5$), speed adjustments ($q_6$), cargo threats ($q_7$), and uncertainty levels ($q_8$). For integration into our causal

modeling framework, these continuous scores are then discretized into binary warning indicators: $B_t = [b_1^t, b_2^t, \ldots, b_8^t]^\top \in \{0,1\}^8$, where $b_i^t = \mathbb{I}\{q_i^t > \tau_i\}$, with threshold $\tau_i$ determined empirically for each dimension. This binary representation enables our neural modules to reason about warning impacts as discrete treatment variables suitable for causal inference. In our design, we view $B_t$ as the actionable "knob" that operators can adjust (e.g., severity, duration, spatial extent), whereas $A_t$ is raw, unstructured text used only to construct $B_t$; all counterfactual questions in this work therefore take the form "what if the warning characteristics had been different?" and are formalized as interventions $do(B_t)$. The thresholds $\{\tau_i\}$ are chosen by combining domain guidelines with small validation sweeps, and we empirically observe that model performance is stable over a broad range of $\tau_i$ values; detailed robustness results are reported in Appendix A.3.

**Spatial-Temporal Embedding.** To capture node-specific contextual information, we incorporate spatial-temporal embeddings that encode both geographical characteristics and temporal patterns. The spatial component captures route coordinates and network topology:

$$\mathbf{E}_i^{(s)} = f_\phi([\mathbf{p}_i, \mathbf{c}_i]) \in \mathbb{R}^{d_s} \tag{1}$$

where $\mathbf{p}_i$ represents geographical coordinates and $\mathbf{c}_i$ contains connectivity features. The temporal component encodes cyclical patterns:

$$\mathbf{E}_t^{(t)} = f_\psi([\mathbf{h}_{\text{hour}}, \mathbf{d}_{\text{day}}, \mathbf{m}_{\text{month}}]) \in \mathbb{R}^{d_t} \tag{2}$$

The final embedding combines both components: $\mathbf{E}_{i,t} = f_\xi([\mathbf{E}_i^{(s)}, \mathbf{E}_t^{(t)}]) \in \mathbb{R}^{d_e}$. Because these embeddings are constructed entirely from information available *before* a warning is issued, they act as additional pre-treatment covariates in our causal adjustment set together with historical flows $\mathbf{X}$ and network structure $\mathbf{V}$.

**Causal Deconfounder Block.** Predicting maritime warning impacts requires distinguishing genuine effects from spurious correlations—a fundamental challenge for traditional forecasting methods Pearl et al. (2000); Glymour et al. (2016). During Typhoon Nanmadol in September 2022, correlation-based models incorrectly attributed Busan port congestion to local warnings, missing the true cause—vessel diversions from Japanese ports creating cascading effects Wang et al. (2021).

Our Deconfounder Block implements a neural approximation of causal adjustment to address this limitation. We adopt the following causal semantics: (i) $A$ denotes the raw warning text (unstructured, non-manipulable), (ii) $B$ is its structured, quantitative representation (our operational treatment), and (iii) $(\mathbf{X}, \mathbf{V}, \mathbf{E})$ are genuinely pre-treatment contextual covariates/confounders that affect both warning impacts and future flows. Given the causal structure in Figure 4, where warning events $A$ generate impact scores $B \in \{0,1\}^8$ that influence future flows $\mathbf{Y}_{t+1:t+T'} \in \mathbb{R}^{N \times T'}$ through pre-treatment contextual covariates/confounders $\mathbf{M} = \{\mathbf{X}, \mathbf{V}\}$ and spatial-temporal embeddings $\mathbf{E}$, where $\mathbf{X}$ represents historical flow patterns and $\mathbf{V}$ captures spatial context, we implement a neural back-door adjustment strategy to control for confounding:

$$P(\mathbf{Y} \mid \text{do}(B)) = \sum_{\mathbf{m}, \mathbf{e}} P(\mathbf{Y} \mid B, \mathbf{M} = \mathbf{m}, \mathbf{E} = \mathbf{e}) \cdot P(\mathbf{M} = \mathbf{m}, \mathbf{E} = \mathbf{e}) \tag{3}$$

This back-door adjustment controls for confounding by conditioning on the sufficient set of pre-treatment covariates $\mathbf{M} = \{\mathbf{X}, \mathbf{V}\}$ and spatial-temporal embeddings $\mathbf{E}$, then marginalizing over their joint distribution, following Pearl's causal hierarchy. Intuitively, $P(\mathbf{Y} \mid \text{do}(B))$ answers policy-relevant questions such as "what if the same maritime context $(\mathbf{X}, \mathbf{V}, \mathbf{E})$ had been exposed to a more/less severe or spatially extensive warning $B$?", while $A$ merely serves as a noisy text source from which $B$ is constructed.

The warning encoder processes impact scores: $\mathbf{h}_B = f_\theta^{(B)}(B_t) \in \mathbb{R}^{d_h}$, where $f_\theta^{(B)} : \{0,1\}^8 \to \mathbb{R}^{d_h}$ is a multi-layer perceptron. The pre-treatment covariates $\mathbf{M}$ are modeled through dedicated encoders:

$$\mathbf{H_X} = f_\phi^{(X)}(\mathbf{X}_{t-T:t}), \quad \mathbf{H_V} = f_\psi^{(V)}(\mathbf{A}_{adj}, \mathbf{X}_t), \quad \mathbf{H_E} = f_\omega^{(E)}(\mathbf{E}_{t-T:t}) \in \mathbb{R}^{N \times d_h} \tag{4}$$

where $f_\phi^{(X)}$ captures historical flow patterns, $f_\psi^{(V)}$ models spatial context through network adjacency $\mathbf{A}_{adj} \in \{0,1\}^{N \times N}$, and $f_\omega^{(E)}$ encodes spatial-temporal embedding features. The deconfounder learns to weight the influence of confounding mediators conditional on warning signals.

We broadcast the warning representation: $\mathbf{H}_B = \mathbf{1}_N \otimes \mathbf{h}_B \in \mathbb{R}^{N \times d_h}$ and compute interaction terms:

$$\mathbf{C}_{BX} = \sigma(\mathbf{W}_{BX}[\mathbf{H}_B, \mathbf{H_X}]), \quad \mathbf{C}_{BV} = \sigma(\mathbf{W}_{BV}[\mathbf{H}_B, \mathbf{H_V}]), \quad \mathbf{C}_{BE} = \sigma(\mathbf{W}_{BE}[\mathbf{H}_B, \mathbf{H_E}])$$

where $[\cdot, \cdot]$ denotes concatenation, and $\{\mathbf{W}_{BX}, \mathbf{W}_{BV}, \mathbf{W}_{BE}\} \subset \mathbb{R}^{d_h \times 2d_h}$ are learnable transformation matrices. The final deconfounded representation combines these weighted interactions:

$$\mathbf{Z}_{\text{causal}} = \mathbf{C}_{BX} \odot \mathbf{H_X} + \mathbf{C}_{BV} \odot \mathbf{H_V} + \mathbf{C}_{BE} \odot \mathbf{H_E} \tag{5}$$

where $\odot$ denotes element-wise multiplication. This yields a neural analogue of back-door adjustment: the interaction gates $(\mathbf{C}_{BX}, \mathbf{C}_{BV}, \mathbf{C}_{BE})$ down-weight spurious correlations between $B$ and $\mathbf{Y}$ that are explainable by $(\mathbf{X}, \mathbf{V}, \mathbf{E})$, while preserving components that correspond to genuine treatment effects.

**ODE-Based Ripple Propagation.** Maritime warnings exhibit propagation patterns resembling physical diffusion processes, yet with domain-specific constraints unique to shipping operations. We model warning propagation as a continuous-time dynamical system governed by neural ordinary differential equations. The temporal evolution of route state vectors follows a coupled system:

$$\frac{d\mathbf{x}_i(t)}{dt} = \underbrace{-\sigma_i(t)\mathbf{x}_i(t)}_{\text{Local decay}} + \underbrace{\sum_{j \in \mathcal{N}(i)} \rho_{ij}(t)\mathbf{R}_{ij}(t)g(\mathbf{x}_j(t), \Delta_{ij})}_{\text{Network coupling}} \tag{6}$$

where $\mathbf{x}_i(t) \in \mathbb{R}^{d_s}$ represents the state vector of route $i$, $\mathcal{N}(i) = \{j : \mathbf{A}_{adj}[i, j] = 1\}$ denotes the neighborhood, and $d_s$ is the state dimension. The system is initialized using the causal representation: $\mathbf{x}_i(0) = g_\xi([\mathbf{Z}_{\text{causal}}[i], \mathbf{e}_i]) \in \mathbb{R}^{d_s}$, where $g_\xi : \mathbb{R}^{d_h + d_e} \to \mathbb{R}^{d_s}$ is an MLP, $\mathbf{e}_i \in \mathbb{R}^{d_e}$ represents learnable route embeddings, and $[\cdot, \cdot]$ denotes concatenation.

The coupling matrix incorporates both geographical and topological distances:

$$\mathbf{R}_{ij}(t) = \exp\left(-\frac{\|\mathbf{p}_i - \mathbf{p}_j\|_2^2}{2\sigma_d^2}\right)\exp(-\lambda h_{ij})\mathbf{I}_{d_s} \tag{7}$$

where $\mathbf{p}_i \in \mathbb{R}^2$ are geographical coordinates, $h_{ij}$ is the shortest path length, $\sigma_d > 0$ controls spatial decay, $\lambda > 0$ is the network decay parameter, and $\mathbf{I}_{d_s}$ is the identity matrix. Note that $\mathbf{R}_{ij}(t)$ and $\rho_{ij}(t)$ play distinct roles: $\mathbf{R}_{ij}(t)$ provides a fixed geometric–topological prior specifying *where* propagation can occur and its baseline strength, whereas $\rho_{ij}(t)$ provides a dynamic, data-driven scaling that determines *how strongly* it propagates under current conditions.

The local decay rate adapts to operational conditions: $\sigma_i(t) = \sigma_0 + \sum_{k=1}^K \sigma_k \phi_k(\mathbf{s}_i(t))$, where $\sigma_0 > 0$ is the baseline rate, $\{\sigma_k\}_{k=1}^K$ are learnable coefficients, and $\phi_k(\mathbf{s}_i(t))$ are feature functions of operational state $\mathbf{s}_i(t) \in \mathbb{R}^{d_{\text{op}}}$ including capacity utilization and weather conditions.

The propagation rate depends on network properties: $\rho_{ij}(t) = \rho_0 + \rho_1 \mathbf{A}_{adj}[i, j] + \rho_2 F_{ij}(t - 1)$, where $\{\rho_0, \rho_1, \rho_2\} \subset \mathbb{R}^+$ are learnable parameters and $F_{ij}(t - 1) \geq 0$ captures historical flow intensity.

Maritime operations exhibit persistent effects due to scheduling constraints. We model this through a dual-exponential decay kernel:

$$g(\mathbf{x}_j, \Delta_{ij}) = \mathbf{x}_j \odot \left[\alpha e^{-\beta\Delta_{ij}} + (1 - \alpha)e^{-\gamma\Delta_{ij}^2}\right] \tag{8}$$

where $\alpha \in [0, 1]$ balances immediate versus persistent effects, $\{\beta, \gamma\} \subset \mathbb{R}^+$ control decay rates, and $\Delta_{ij} = \|\mathbf{p}_i - \mathbf{p}_j\|_2/v_{\text{avg}}$ represents transit time with average vessel speed $v_{\text{avg}}$. The ODE system is solved using adaptive Runge-Kutta methods over interval $[0, T_{\text{int}}]$ where $T_{\text{int}} = T' \cdot \Delta t$: $\mathbf{H}^{\text{ripple}}[i, :] = \mathbf{x}_i(T_{\text{int}})^T$ for $i = 1, 2, \ldots, N$ where $\mathbf{H}^{\text{ripple}} \in \mathbb{R}^{N \times d_s}$. In practice we instantiate this ODE with a Dormand–Prince RK45 solver (via `torchdiffeq`), which provides adaptive step sizes and a good accuracy–efficiency trade-off for heterogeneous maritime dynamics. From a causal perspective, this continuous-time propagation acts as an explicit interference model: a treatment applied on one route (through its contribution to $\mathbf{Z}_{\text{causal}}$) can influence neighboring routes over time along the graph edges, with the decay terms controlling how far and how long warning-induced perturbations persist.

**Attention-Based Integration.** Following ODE-based ripple propagation, we employ multi-head self-attention to capture strategic dependencies beyond physical diffusion patterns. Maritime operations involve coordinated decision-making that creates non-local interdependencies complementing diffusion-based propagation. The attention mechanism operates on concatenated representations: $\mathbf{H}_{\text{fused}} = h_\zeta([\mathbf{Z}_{\text{causal}}, \mathbf{H}^{\text{ripple}}]) \in \mathbb{R}^{N \times d_h}$, where $h_\zeta : \mathbb{R}^{N \times (d_h + d_s)} \to \mathbb{R}^{N \times d_h}$ is a projection network. Multi-head attention captures diverse dependency patterns:

$$\mathbf{H}_{\text{attn}} = \text{MultiHead}(\mathbf{H}_{\text{fused}}, \mathbf{H}_{\text{fused}}, \mathbf{H}_{\text{fused}}) \in \mathbb{R}^{N \times d_h}.$$

Final predictions are generated through temporal projection: $\mathbf{Y}_{t+1:t+T'} = f_{\text{out}}(\mathbf{H}_{\text{attn}}) \in \mathbb{R}^{N \times T'}$, where $f_{\text{out}} : \mathbb{R}^{N \times d_h} \to \mathbb{R}^{N \times T'}$ maps representations to future flows. The model is trained using regularized mean squared error:

$$\mathcal{L}(\mathbf{\Theta}) = \frac{1}{2NT'} \left\| \mathbf{Y}_{t+1:t+T'} - \hat{\mathbf{Y}}_{t+1:t+T'} \right\|_F^2 + \frac{\lambda}{2} \|\mathbf{\Theta}\|_2^2 \tag{9}$$

where $\mathbf{\Theta}$ represents all learnable parameters and $\lambda > 0$ is the regularization coefficient.

## 5 Experiments

### 5.1 Datasets & Experimental Settings

We conduct experiments on two of the most active maritime regions to evaluate the effectiveness of our proposed method: the East Asia and Northwest Europe maritime flow networks. Both networks are constructed using Spire AIS data from 2022 and port information from Lloyd's List Intelligence. Specifically, we perform port-to-port trajectory segmentation and aggregate departure–arrival records into 12-hour intervals (two frames per day) over a full year, ensuring both temporal granularity and seasonal coverage. In addition, we collect public weather data from the Global Forecast System (GFS) and a publicly available maritime warning dataset. Both are used to generate warning score vectors through a prompt-based large language model (LLM) approach. Specifically, we employ the OpenAI ChatGPT-4o API to transform maritime event sequences into structured and multi-dimensional warning vectors.

Our flow forecasting task is defined as predicting the next 4 time steps based on the preceding 8. We compare our method against one classical statistical baseline (Historical Average) and eight state-of-the-art spatio-temporal graph neural networks: STGCN Yu et al. (2018), MTGNN Wu et al. (2020), GMAN Zheng et al. (2020), GraphWaveNet Wu et al. (2019), AGCRN Bai et al. (2020), PDFormer Jiang et al. (2023), STAEformer Liu et al. (2023), and CaST Xia et al. (2023). To assess the contributions of individual components, we conduct an ablation study with three variants of our framework: (1) RippleNet-DEF, which removes the neural back-door adjustment block, thereby disabling the causal disentanglement mechanism; (2) RippleNet-ODE, which omits the ODE-based ripple propagation module, thus disabling continuous-time dynamics modeling; (3) RippleNet-LLM, which replaces the LLM-generated warning vectors with one-dimensional binary indicators to represent the warning event status (i.e., 0 = no event; 1 = event). Model performance is evaluated using Mean Absolute Error (MAE) and Root Mean Squared Error (RMSE). More details about the datasets and experiments of our model are provided in Appendix A.1.

### 5.2 Experimental Results

**Overall Performance.** As shown in Table 1, our experiments validate the core hypothesis: traditional correlation-based methods fail under warning-induced cascading effects, while our RippleNet achieves superior performance across both maritime regions. In East Asia, RippleNet outperforms the best baseline PDFormer by 9.5% MAE and 2.5% RMSE at 12-hour predictions, maintaining consistent advantages even at 48-hour horizons (6.9% and 1.3% improvements respectively). This validates our assertion that warning impacts exhibit persistent propagation requiring causal mechanisms for accurate modeling. Northwest Europe results are more striking, with RippleNet achieving 19.6% MAE and 7.4% RMSE improvements over MTGNN. This substantial gap reflects the challenge of modeling long-range causal propagation in sparse networks, where our ODE-based ripple module effectively captures cross-regional influences that traditional methods miss. Crucially, all deep learning baselines exhibit limitations under warning scenarios. Despite strong normal-condition performance, methods like PDFormer and STAEformer rely on historical correlations and

Table 1: 5-run average flow prediction performance comparison on East Asia and Northwest Europe maritime transportation networks on different time horizons. **Bold** (red) denotes the best overall result, and underline (orange) denotes the best baseline excluding our proposed RippleNet.

| Dataset | | Metric | HA | STGCN | MTGNN | AGCRN | GMAN | GraphWaveNet | PDFormer | STAEformer | CaST | RippleNet | Improve |
|---|---|---|---|---|---|---|---|---|---|---|---|---|---|
| East Asia | 12 hours | MAE | 2.819 | 2.370 | 2.444 | 2.475 | 2.337 | 2.325 | 2.144 | 2.196 | 2.232 | **1.940** | **9.5%** ↑ |
| | | RMSE | 6.915 | 6.333 | 6.511 | 6.383 | 6.371 | 6.343 | 6.020 | 6.242 | 6.287 | **5.872** | **2.5%** ↑ |
| | 24 hours | MAE | 2.818 | 2.469 | 2.564 | 2.524 | 2.434 | 2.429 | 2.230 | 2.283 | 2.306 | **2.027** | **9.1%** ↑ |
| | | RMSE | 6.915 | 6.350 | 6.528 | 6.689 | 6.417 | 6.412 | 6.197 | 6.227 | 6.290 | **5.988** | **3.4%** ↑ |
| | 36 hours | MAE | 2.819 | 2.654 | 2.665 | 2.588 | 2.590 | 2.563 | 2.300 | 2.383 | 2.412 | **2.137** | **7.1%** ↑ |
| | | RMSE | 6.912 | 6.504 | 6.502 | 6.432 | 6.436 | 6.418 | 6.247 | 6.315 | 6.329 | **6.035** | **3.4%** ↑ |
| | 48 hours | MAE | 2.818 | 2.694 | 2.664 | 2.593 | 2.624 | 2.607 | 2.321 | 2.400 | 2.404 | **2.162** | **6.9%** ↑ |
| | | RMSE | 6.910 | 6.724 | 6.703 | 6.447 | 6.503 | 6.448 | 6.270 | 6.365 | 6.368 | **6.186** | **1.3%** ↑ |
| Northwest Europe | 12 hours | MAE | 1.658 | 1.259 | 1.228 | 1.323 | 1.338 | 1.316 | 1.248 | 1.302 | 1.265 | **0.987** | **19.6%** ↑ |
| | | RMSE | 3.185 | 2.743 | 2.741 | 2.860 | 2.879 | 2.851 | 2.715 | 2.812 | 2.738 | **2.513** | **7.4%** ↑ |
| | 24 hours | MAE | 1.656 | 1.350 | 1.257 | 1.329 | 1.355 | 1.330 | 1.286 | 1.337 | 1.314 | **1.014** | **19.3%** ↑ |
| | | RMSE | 3.181 | 2.957 | 2.762 | 2.868 | 2.932 | 2.875 | 2.820 | 2.937 | 2.894 | **2.529** | **8.4%** ↑ |
| | 36 hours | MAE | 1.658 | 1.460 | 1.288 | 1.364 | 1.378 | 1.369 | 1.326 | 1.392 | 1.350 | **1.050** | **18.5%** ↑ |
| | | RMSE | 3.189 | 3.123 | 2.884 | 2.939 | 2.950 | 2.946 | 2.890 | 3.061 | 3.027 | **2.581** | **10.5%** ↑ |
| | 48 hours | MAE | 1.665 | 1.488 | 1.304 | 1.385 | 1.387 | 1.383 | 1.334 | 1.404 | 1.392 | **1.175** | **9.9%** ↑ |
| | | RMSE | 3.205 | 3.142 | 2.886 | 2.967 | 2.974 | 2.969 | 2.940 | 3.069 | 3.053 | **2.798** | **3.0%** ↑ |

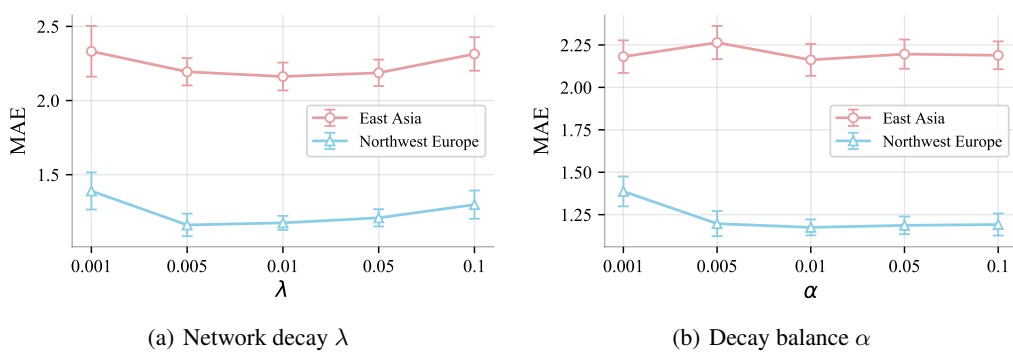

(a) Network decay $\lambda$  (b) Decay balance $\alpha$

Figure 5: Hyper-parameter analysis (mean ± std) for 48-hour prediction horizon.

cannot adapt to causal structure changes. RippleNet's learned deconfounder explicitly disentangles genuine causal effects from spurious correlations, maintaining stability during anomalous events and demonstrating the necessity of causal reasoning for robust maritime prediction systems.

**Ablation Study.** Table 2 demonstrates each component's critical contribution through systematic removal. Eliminating the neural backdoor adjustment (RippleNet-DEF) increases MAE by 10.7%/18.7% across regions, validating that explicit causal disentanglement is essential for distinguishing genuine effects from confounding factors during warning events. Removing ODE-based ripple propagation (RippleNet-ODE) causes more severe degradation with 16.4%/29.5% MAE increases, confirming that continuous-time dynamics are

Table 2: Ablation study on MAE comparison over both maritime transportation networks.

| DEF | ODE | LLM | Datasets | Time Horizon | | | |
|---|---|---|---|---|---|---|---|
| | | | | 12 hours | 24 hours | 36 hours | 48 hours |
| ✗ | ✓ | ✓ | East Asia | 2.147 | 2.325 | 2.348 | 2.354 |
| | | | Northwest Europe | 1.172 | 1.261 | 1.336 | 1.245 |
| ✓ | ✗ | ✓ | East Asia | 2.258 | 2.396 | 2.393 | 2.407 |
| | | | Northwest Europe | 1.278 | 1.316 | 1.344 | 1.390 |
| ✓ | ✓ | ✗ | East Asia | 2.092 | 2.156 | 2.249 | 2.278 |
| | | | Northwest Europe | 1.024 | 1.043 | 1.065 | 1.194 |
| ✓ | ✓ | ✓ | East Asia | **1.940** | **2.027** | **2.137** | **2.162** |
| | | | Northwest Europe | **0.987** | **1.014** | **1.050** | **1.175** |

fundamental for modeling warning propagation through maritime networks. The LLM-generated warning vectors exhibit moderate yet consistent importance compared to RippleNet-LLM, with 7.8% and 3.7% performance improvements, underscoring the value of structured warning quantification over event treatment representation.

**Hyper-parameter Analysis.** Figure 5 examines sensitivity to key hyperparameters. The network decay parameter $\lambda$ achieves optimal performance around 0.01, balancing propagation reach with noise reduction. Lower values cause excessive propagation, while higher values limit long-range dependency capture. The decay balance parameter $\alpha$ also performs best at 0.01, indicating maritime

warnings exhibit combined immediate and persistent effects. This validates our dual-exponential formulation for modeling complex temporal dynamics in warning propagation.

## 5.3 CASE STUDY

To further examine RippleNet's interpretability in capturing warning-induced causal impacts, we present two case studies focused on the ripple effects triggered by Typhoon Nanmadol in September 2022. Figure 6(a) first shows the temporal dynamics of LLM-derived warning vectors averaged over affected routes, with spatial impact (Q1) and rerouting need (Q3) peaking around typhoon landfall. Delay risk (Q2) and cargo threat (Q6) remain low except during peak disruption, reflecting route-level resilience. Figure 6(b) maps average scores across four routes at varying distances from the typhoon's center; closer routes exhibit higher spatial impact, congestion risk, and rerouting need, while even distant ones show moderate uncertainty (Q8) and duration impact (Q4), indicating indirect ripple effects. In addition, Figure 6(a) shows a clear lead and lag pattern: Q1 and Q3 rise first, port congestion (Q5) responds later, and duration (Q4) remains flat, while Figure 6(b) reveals a monotonic distance gradient in which distant routes retain elevated Q8 and Q4, indicating spillovers mediated by the network. These results demonstrate that RippleNet's warning scoring yields semantically meaningful signals that adapt to both spatial and operational contexts, supporting robust causal modeling.

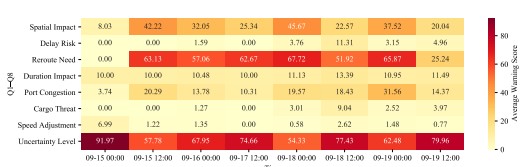

(a) Temporal evolution of average warning scores over time across eight dimensions (Q1–Q8) during Typhoon Nanmadol (September 15–19, 2022).

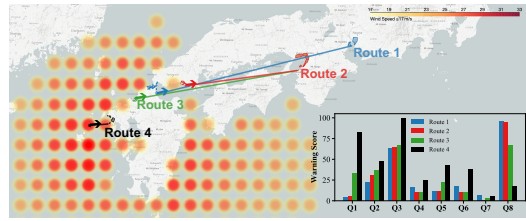

(b) Spatial comparison of warning scores across four routes at varying distances from the typhoon center.

Figure 6: LLM-based warning score analysis.

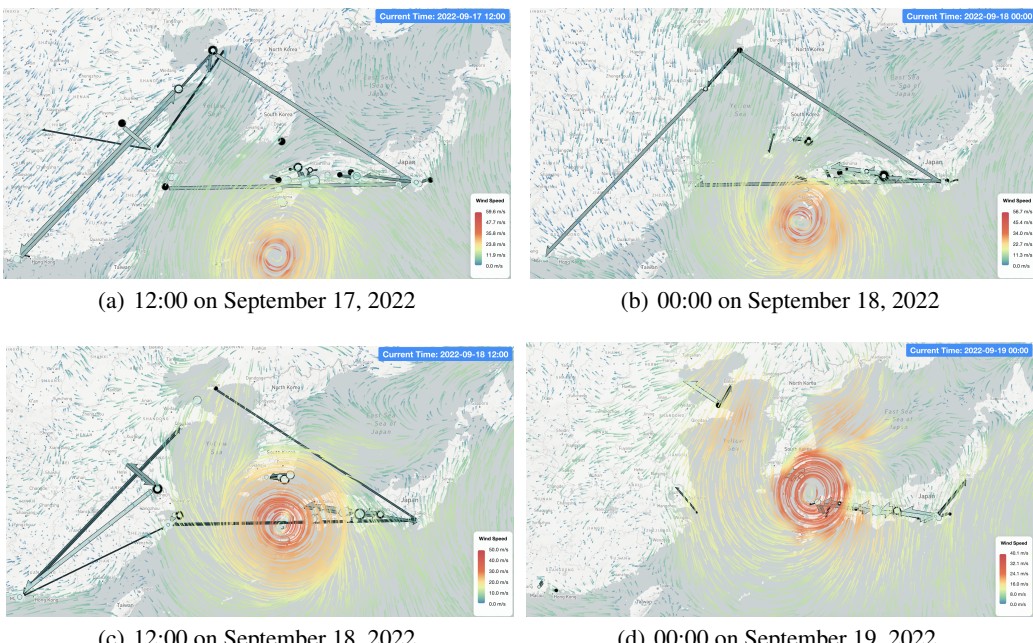

(a) 12:00 on September 17, 2022

(b) 00:00 on September 18, 2022

(c) 12:00 on September 18, 2022

(d) 00:00 on September 19, 2022

Figure 7: Top-50 Negative Causal Effects at each time step during Typhoon Nanmadol.

Moreover, to interpret the temporal suppression of maritime flow by warning signals, we visualize the top 50 strongest negative causal effects at each time step during Typhoon Nanmadol, as identified by RippleNet. At 12:00 on September 17 (Figure 7(a)), suppressive effects concentrate along Japan's southern coast, where initial warnings emerge. As the typhoon progresses, these effects intensify and shift northeastward, peaking around 00:00 on September 18 (Figure 7(b)). By 12:00 on September 18 (Figure 7(c)), they reach the East China Sea and Korean waters despite mild local weather, indicating long-range propagation driven by operational and topological factors. At 00:00 on September 19 (Figure 7(d)), the vortex is more compact and the wind field more coherent, and routes within the warning envelope appear more clustered and visibly constrained along the Japanese coastline, indicating a localized consolidation of suppressive effects rather than further network-wide spread. Unlike correlation-based attention, our model offers interpretable and magnitude-aware representations of ripple effects, enhancing maritime decision-making under warning conditions.

## 6 CONCLUSION

We presented RippleNet, a novel framework that bridges causal inference with spatio-temporal prediction for maritime networks under warnings. By incorporating learned deconfounder and continuous-time dynamics, our approach successfully disentangles causal effects from correlations—a fundamental limitation of existing methods. The ODE-based ripple propagation module captures both immediate and lingering effects through adaptive decay functions, while LLM-generated warning vectors enable systematic quantification of heterogeneous disruptions. Empirical results validate our theoretical insights: causal-aware architectures significantly outperform correlation-based models when historical patterns are violated by anomalous events. Future work includes extending to multi-modal transportation networks and developing uncertainty quantification methods for operational deployment.

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

# A APPENDIX

## A.1 DATASETS AND EXPERIMENTAL SETTINGS

Table 1: Statistics of datasets.

| Dataset | Num of Ports | Num of Port Clusters | Num of Ship Routes |
|---|---|---|---|
| East Asia | 555 | 196 | 689 |
| Northwest Europe | 812 | 264 | 395 |

**Data Preparation.** We conduct experiments on two of the most active maritime regions to evaluate the effectiveness of our proposed method: the East Asia and Northwest Europe maritime flow networks. The East Asia network includes Japan, the Republic of Korea, and China, while the Northwest Europe network covers the Netherlands, the United Kingdom of Great Britain and Northern Ireland, Belgium, Denmark, Germany, France, Norway, Sweden, and Finland. Both maritime transportation networks are constructed using 2022 AIS data from Spire[1], which includes cargo, tanker, and passenger vessels, as well as port location data from Lloyd's List Intelligence[2]. Trajectory segmentation is first performed based on zero-speed thresholds and spatial constraints that determine whether a vessel is located within a port boundary. To reduce the impact of short-range trajectory noise, port locations are clustered using a 0.05-degree threshold during flow aggregation. To capture inter-port connectivity, we construct directed adjacency graphs in which each edge represents a flow from a departure port to an arrival port within a 12-hour temporal aggregation window (i.e., two frames per day) over the full year of 2022. This directed graph formulation reflects the inherent asymmetry of real-world maritime flows, where traffic between port pairs may vary significantly in volume and direction. Table 1 summarizes key network statistics, and Figure 1 provides a global visualization of the resulting maritime flow structure. As shown in Figure 2, the Northwest Europe network exhibits relatively sparse connectivity, with many directed links connecting only isolated port pairs, suggesting a more fragmented or specialized shipping structure compared to the denser and more interconnected network observed in East Asia. In addition, we incorporate public weather data from the Global Forecast System (GFS) and maritime warning bulletins from the Japan Coast Guard[3] for East Asia and from the NAVAREA I, II, and XIX coordinators (UK Hydrographic Office, SHOM, and the Norwegian Coastal Administration)[4] for Northwest Europe, all of which are transformed into structured warning score vectors using our prompt-based method. The weather variable considered in this study is wind speed, with a spatial resolution of 0.5 degrees and the same 12-hour temporal granularity as the flow data. Strong wind events—defined as those with speeds exceeding 17 m/s—are extracted and converted into structured warning indicators, as illustrated in Figure 3 in the main text. The second warning dataset is processed in a similar fashion to ensure consistency across all warning inputs.

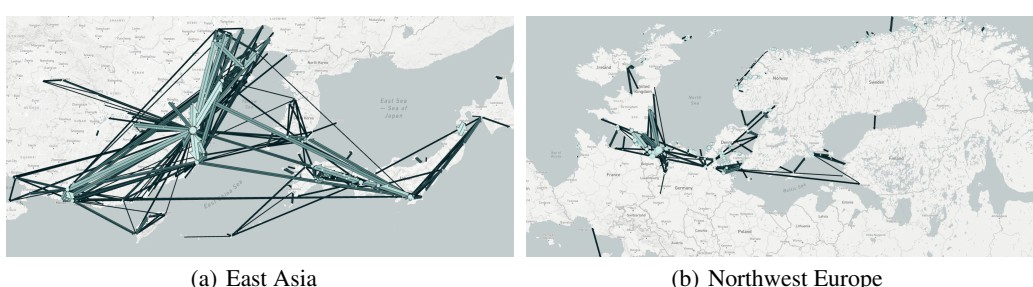

(a) East Asia           (b) Northwest Europe

Figure 1: Maritime transportation network for both two target areas.

---

[1] https://spire.com/maritime/
[2] https://www.lloydmaritime.com/en/module/port-management
[3] https://www1.kaiho.mlit.go.jp/TUHO/weekly/weekly_en.html
[4] See the official MSI services of the UK Hydrographic Office (NAVAREA I), SHOM (NAVAREA II), and the Norwegian Coastal Administration (NAVAREA XIX).

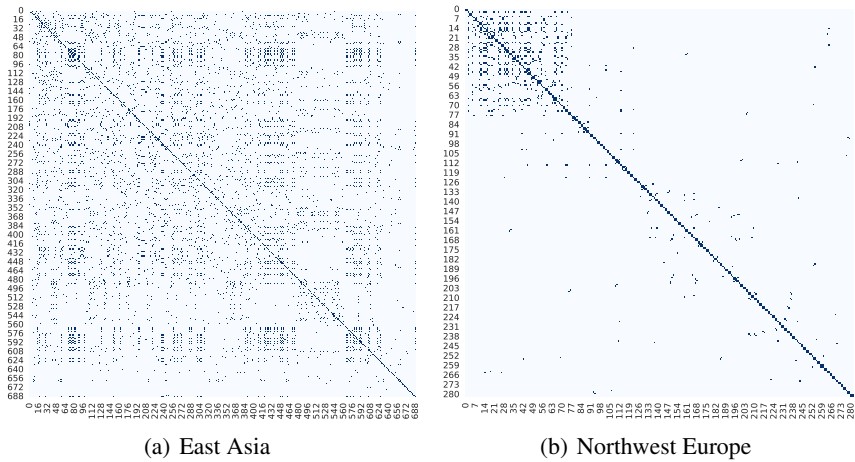

(a) East Asia      (b) Northwest Europe

Figure 2: The heatmap of the adjacency matrix.

**Code Reproducibility.** Our code is publicly available in the anonymous supplementary material.

**Baseline Methods.** We compare our approach against one classical statistical method and eight recent spatio-temporal graph neural networks as baselines for the task of maritime flow prediction:

- **Historical Average (HA)**: A conventional time-series baseline that predicts future values by averaging historical observations at corresponding time-of-day and day-of-week slots.

- **STGCN** Yu et al. (2018): This model applies alternating gated temporal convolutional blocks and spatial graph convolution layers to jointly capture temporal dynamics and spatial structure in graph data.

- **MTGNN** Wu et al. (2020): A model designed for multivariate time-series forecasting, it learns a directed graph structure via a graph-learning layer, then applies graph convolution combined with dilated temporal convolution to model adaptive spatial and long-range temporal dependencies in an end-to-end framework.

- **AGCRN** Bai et al. (2020): This model embeds node-specific adaptive graph convolution inside gated recurrent units to model heterogeneous node dynamics and long-term temporal dependencies in spatio-temporal traffic data.

- **GMAN** Zheng et al. (2020): This encoder–decoder graph model stacks spatio-temporal attention blocks and uses a transform-attention bridge to align past and future steps for stable multi-horizon forecasts.

- **GraphWaveNet** Wu et al. (2019): This model learns an adaptive adjacency and applies dilated causal convolutions to capture hidden spatial links and long-range temporal patterns.

- **PDFormer** Jiang et al. (2023): A transformer-based architecture that integrates progressive trend-seasonal decomposition with spatial attention, designed to enhance representation of both local and global spatio-temporal patterns.

- **STAEformer** Liu et al. (2023): A hybrid encoder–decoder transformer that jointly leverages spatial self-attention and temporal modeling at multiple scales, aiming to capture complex interactions in spatio-temporal graphs.

- **CaST** Xia et al. (2023): A causal spatio-temporal transformer framework that separates spatial and temporal convolution paths based on causal structure, improving interpretability and forecasting accuracy in dynamic settings.

**Implementation Details.** We implement RippleNet using PyTorch 2.2.1 and train all models on an NVIDIA RTX 4090D GPU. The model architecture consists of the following components: (i) an input projection layer mapping 4-dimensional raw features to a 24-dimensional hidden space; (ii) temporal embeddings encoding time-of-day and day-of-week information, each projected to a

Table 2: Training speed and GPU memory usage among baselines and our proposed method. Here, efficiency values are reported as East Asia / Northwest Europe.

| Metric \ Model | STGCN | MTGNN | AGCRN | GMAN | GraphWaveNet | PDFormer | STAEformer | CaST | RippleNet |
|---|---|---|---|---|---|---|---|---|---|
| Time per epoch (s) | 0.84 / 0.59 | 1.82 / 1.62 | 3.36 / 3.18 | 5.39 / 2.49 | 3.51 / 1.64 | 8.36 / 3.80 | 246.61 / 115.23 | 5.97 / 2.83 | 13.85 / 7.79 |
| GPU memory (GB) | 0.11 / 0.07 | 0.32 / 0.19 | 2.17 / 1.21 | 2.83 / 1.68 | 1.58 / 0.94 | 4.17 / 2.50 | 15.51 / 9.08 | 3.56 / 2.39 | 7.15 / 4.74 |

24-dimensional space; (iii) adaptive warning embeddings of shape $(8, 178, 24)$ to capture localized causal signals; (iv) a back-door causal module with a hidden dimension of 16; and (v) a 3-layer causal attention mechanism with 4 heads per layer. In addition, the ODE-based causal dynamics module uses adaptive Runge–Kutta solvers with 5 causal state variables. To support structured intervention modeling, we binarize each warning vector $[Q_1, \dots, Q_8] \in [0, 100]^8$ into $B \in \{0, 1\}^8$ using dimension-specific thresholds $\tau_i$. The threshold values are set as $\tau_i = [30, 40, 40, 40, 60, 60, 30, 70]$, based on empirical percentiles and domain knowledge. For both maritime transportation networks, we split the data strictly in chronological order: the first $60\%$ of timestamps are used for training, the next $10\%$ for validation, and the final $30\%$ are held out for testing. Training is performed using the Adam optimizer with a learning rate of $10^{-3}$, a batch size of 8, and weight decay set to $1.5 \times 10^{-3}$. A multi-step learning rate scheduler reduces the learning rate by a factor of 0.1 at epochs $\{25, 45, 65\}$. Early stopping with a patience of 10 epochs is applied to prevent overfitting. For all baseline models, we start from the authors' recommended configurations and perform small hyperparameter sweeps on the validation set (e.g., hidden dimension and dropout), selecting the configuration with the best validation MAE and reporting test performance under that setting. All experiments are conducted with fixed random seeds to ensure reproducibility.

**Computational Efficiency.** To quantify computational efficiency, we report both the average wall-clock training time per epoch and peak GPU memory usage for all baselines and our proposed RippleNet on the East Asia and Northwest Europe networks in Table 2: classical GNN-based models such as STGCN and GraphWaveNet are the fastest and most lightweight, while transformer-based architectures (PDFormer, STAEformer, CaST) incur noticeably higher computational costs. RippleNet lies in a moderate regime: it is slower and more memory-consuming than the simplest graph convolutional baselines due to the causal deconfounder and ODE modules, but remains substantially more efficient than STAEformer and comparable to other transformer-style models, with per-epoch training time below 14 seconds and peak memory usage under 8 GB on East Asia.

**Causal Semantics and Deconfounder Implementation.** For completeness, we summarize here the causal semantics adopted in the main text and how they are instantiated in our implementation. We distinguish between: (i) the raw warning text $A$ (e.g., meteorological bulletins or security advisories), which is high-dimensional and non-manipulable; (ii) the structured warning vector $B \in \{0, 1\}^8$, which encodes operationally meaningful warning attributes (spatial extent, delay risk, rerouting need, etc.) and serves as our treatment variable; and (iii) the contextual covariates $(\mathbf{X}, \mathbf{V}, \mathbf{E})$, which are genuinely pre-treatment and capture historical flows, network structure, and spatial–temporal embeddings, respectively. In all causal statements, when we write $P(\mathbf{Y} \mid \text{do}(B))$ we are asking "what would future flows have looked like under a counterfactual warning profile $B$ in the same maritime context $(\mathbf{X}, \mathbf{V}, \mathbf{E})$?", while $A$ is only used to construct $B$ via the LLM-based scoring pipeline.

The neural deconfounder block operationalizes the back-door adjustment $P(\mathbf{Y} \mid \text{do}(B)) = \sum_{\mathbf{m}, \mathbf{e}} P(\mathbf{Y} \mid B, \mathbf{M} = \mathbf{m}, \mathbf{E} = \mathbf{e}) P(\mathbf{M} = \mathbf{m}, \mathbf{E} = \mathbf{e})$ in a representation-learning manner. Concretely, we encode $B$ into a warning representation $\mathbf{h}_B$, and $(\mathbf{X}, \mathbf{V}, \mathbf{E})$ into $\mathbf{H_X}, \mathbf{H_V}, \mathbf{H_E}$, then learn interaction gates $(\mathbf{C}_{BX}, \mathbf{C}_{BV}, \mathbf{C}_{BE})$ that modulate how much of each contextual signal remains associated with $B$ after adjustment. The resulting deconfounded representation $\mathbf{Z}_{\text{causal}}$ thus down-weights components of the association between $B$ and $\mathbf{Y}$ that are fully explained by pre-treatment context, while preserving residual variation attributable to genuine warning effects. This matches the causal semantics of our method and avoids misinterpreting $B$ as a mediator of some upstream "true" treatment.

For the ODE-based ripple propagation module, we instantiate the continuous-time dynamics with a Dormand–Prince RK45 solver implemented via `torchdiffeq`, using adaptive step sizes to balance accuracy and efficiency. From a causal perspective, this ODE layer provides an explicit inter-

Table 3: 5-run average flow prediction performance comparison (mean $\pm$ std) on both maritime transportation networks.

| Dataset | Metric | STGCN | MTGNN | AGCRN | GMAN | GraphWaveNet | PDFormer | STAEformer | CaST | RippleNet |
|---------|--------|-------|-------|-------|------|--------------|----------|------------|------|-----------|
| East Asia | MAE | $2.694 \pm 0.031$ | $2.664 \pm 0.039$ | $2.593 \pm 0.056$ | $2.624 \pm 0.075$ | $2.607 \pm 0.082$ | $2.321 \pm 0.035$ | $2.400 \pm 0.118$ | $2.404 \pm 0.193$ | $\mathbf{2.162 \pm 0.096}$ |
| | RMSE | $6.724 \pm 0.085$ | $6.703 \pm 0.096$ | $6.447 \pm 0.143$ | $6.503 \pm 0.186$ | $6.448 \pm 0.205$ | $6.270 \pm 0.094$ | $6.365 \pm 0.298$ | $6.368 \pm 0.373$ | $\mathbf{6.186 \pm 0.255}$ |
| Northwest Europe | MAE | $1.488 \pm 0.014$ | $1.304 \pm 0.018$ | $1.385 \pm 0.037$ | $1.387 \pm 0.048$ | $1.383 \pm 0.059$ | $1.334 \pm 0.020$ | $1.404 \pm 0.061$ | $1.392 \pm 0.095$ | $\mathbf{1.175 \pm 0.047}$ |
| | RMSE | $3.142 \pm 0.037$ | $2.886 \pm 0.024$ | $2.967 \pm 0.090$ | $2.974 \pm 0.119$ | $2.969 \pm 0.136$ | $2.940 \pm 0.054$ | $3.069 \pm 0.174$ | $3.053 \pm 0.128$ | $\mathbf{2.798 \pm 0.093}$ |

ference model: a change in the treatment profile $B$ on one route first alters its local causal representation $\mathbf{Z}_{causal}$, and then propagates along graph edges over continuous time, with spatial decay and dual-exponential temporal kernels controlling how far and how long these warning-induced perturbations influence other routes. This makes the implied interference pattern transparent and separates it from purely correlation-based attention mechanisms.

## A.2 ADDITIONAL QUANTITATIVE EXPERIMENTAL RESULTS

To further validate the effectiveness and robustness of `RippleNet`, we report additional quantitative results in this section, including 5-run mean $\pm$ standard deviation comparisons with baseline models, analyses of causal baselines and treatment variants, and cross-region transfer learning experiments under the 48-hour prediction horizon setting.

**Performance Comparison with Uncertainty.** Table 3 reports the 5-run average performance (mean $\pm$ standard deviation) of all baselines and RippleNet. Using MAE as the primary evaluation metric, paired Student's t-tests over the 5 runs indicate that RippleNet significantly outperforms all baselines on both datasets ($p < 0.01$). The RMSE results follow the same ranking and further reinforce this advantage, confirming that the performance gains are consistent across error metrics rather than an artifact of a particular loss function.

**Causal Baselines and Treatment Variants.** To further isolate the effect of explicit treatment modeling, we compare our proposed RippleNet against two canonical counterfactual baselines—CFR and TarNet—as well as the causal spatio-temporal transformer CaST and two treatment variants of RippleNet. CFR and TarNet are non-graph causal representation models that learn balanced latent spaces for treated and control samples, while CaST is a causal GNN that explicitly separates spatial and temporal paths. In our RippleNet, the treatment is encoded as an 8-dimensional *binary* warning vector with *dimension-specific thresholds* derived from maritime domain knowledge; RippleNet-CT instead feeds the continuous LLM warning scores directly as treatments, and RippleNet-Event collapses all warning information into a single binary event flag (0/1). Table 4 shows that all *causal + treatment-aware* models outperform purely correlation-based GNN baselines. Among the causal baselines, CaST already improves over CFR and TarNet by leveraging graph structure and spatio-temporal reasoning. Our proposed RippleNet further achieves the best performance on both regions: our binary and thresholded 8-dimensional treatment vector provides a clean, intervention-ready representation that filters out noisy low-confidence scores while preserving heterogeneous warning dimensions, whereas RippleNet-CT can be more sensitive to LLM score calibration, and RippleNet-Event discards most of the structured warning semantics by compressing them into a single scalar.

Table 4: Performance comparison of causal baselines and different treatment variants of our method on both maritime transportation networks.

| Dataset | Metric | CFR | TarNet | CaST | RippleNet | RippleNet-CT | RippleNet-Event |
|---------|--------|-----|--------|------|-----------|--------------|-----------------|
| East Asia | MAE | 2.654 | 2.727 | 2.404 | **2.162** | 2.174 | 2.278 |
| | RMSE | 6.630 | 6.845 | 6.368 | **6.186** | 6.203 | 6.229 |
| Northwest Europe | MAE | 1.476 | 1.502 | 1.392 | **1.175** | 1.178 | 1.194 |
| | RMSE | 2.991 | 3.152 | 3.053 | **2.798** | 2.805 | 2.861 |

**Region Transfer Test.** To address the question of whether our framework can generalize to additional regions, we design a cross-region transfer experiment using CaST and our RippleNet. We

consider both directions: (i) East Asia → Northwest Europe and (ii) Northwest Europe → East Asia, which differ markedly in network density, route structure, and warning patterns. For each direction, we first train CaST and RippleNet on the source-domain training split. We then fine-tune all parameters on a small labeled adaptation subset of the target-domain training period, corresponding to the earliest 20% of target-domain timestamps. The best checkpoint is selected based on the target-domain validation set, and the held-out target test split (last 30% of timestamps) is used exclusively for evaluation. Table 5 summarizes the 5-run average transfer performance (mean ± std). In both directions, RippleNet achieves lower MAE and RMSE than CaST, with noticeably smaller variance across runs. This suggests that explicitly modeling warning-induced ripple effects in continuous time not only improves in-domain accuracy, but also enhances robustness and generalization when transferring between structurally different maritime regions.

Table 5: Transfer learning performance of CaST and RippleNet between the East Asia and Northwest Europe maritime transportation networks (5-run mean ± std).

| Source Dataset | Target Dataset | Metric | CaST | RippleNet |
|---|---|---|---|---|
| East Asia | Northwest Europe | MAE | $1.528 \pm 0.254$ | $\mathbf{1.427 \pm 0.143}$ |
| | | RMSE | $3.350 \pm 0.592$ | $\mathbf{3.169 \pm 0.353}$ |
| Northwest Europe | East Asia | MAE | $2.796 \pm 0.248$ | $\mathbf{2.649 \pm 0.191}$ |
| | | RMSE | $6.854 \pm 0.480$ | $\mathbf{6.518 \pm 0.416}$ |

## A.3 PROMPT-BASED WARNING VECTORS

**Prompt Designs.** Here, we present a collection of prompts designed to infer warning score vectors from structured maritime warning information. These score vectors are subsequently used as treatment variables within our modeling framework—RippleNet.

---
**Prompt-based Maritime Warning Vector Generation**

*You are an AI assistant specialized in maritime operations. Given structured maritime warning information and a set of candidate shipping routes, your task is to:*
*1. Identify which shipping routes are possibly affected by the warning.*
*2. For each affected route, score the level of impact for the following 8 aspects on a scale from 0 to 100 (higher values indicate stronger and more certain impact). The scoring should reflect the **impact of the warning on the route**.*
*Please score based on the following criteria:*
***Q1. Spatial Impact** — The geographical extent to which the warning affects this route, including both direct coverage and surrounding influence.*
***Q2. Delay Risk** — The level of schedule disruption this warning is likely to cause for vessels on this route.*
***Q3. Reroute Need** — The degree to which this warning creates a need to reroute or avoid this route due to operational or safety concerns.*
***Q4. Duration Impact** — The expected increase in overall travel or operation time for this route caused by the warning.*
***Q5. Port Congestion** — The extent to which the warning contributes to congestion or reduced throughput at ports along this route.*
***Q6. Cargo Threat** — The severity of risk posed by the warning to cargo safety, security, or condition on this route.*
***Q7. Speed Adjustment** — The level of speed alteration expected on this route (e.g., slow steaming or acceleration) due to the warning.*
***Q8. Uncertainty Level** — The degree of ambiguity or lack of reliable information in the warning's implications for this route.*
---

**Rationality Analysis.** To assess the reliability of the warning scores generated by the large language model (LLM), we conduct a rationality analysis based on the temporal patterns shown in

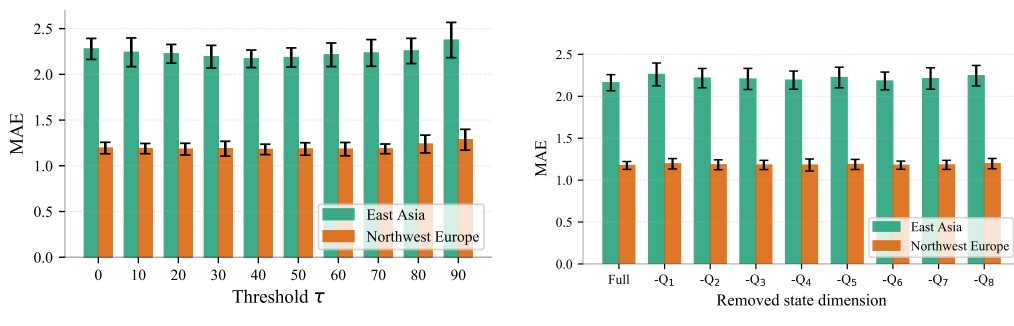

(a) Effect of warning threshold $\tau$          (b) Effect of removing warning dimensions $Q_1$–$Q_8$

Figure 3: Robustness analysis (mean $\pm$ std) of Prompt-based warning vectors.

Figure 6(a) in the main text. This analysis aims to evaluate whether the LLM-derived warning scores are consistent with real-world typhoon dynamics and maritime operational responses.

**Q1:** The first peak occurs at 09-15 12:00 ($\approx 42\%$), corresponding to Typhoon Nanmadol's initial approach to the route area. A second peak appears at 09-18 00:00 ($\approx 45\%$), aligning with the typhoon's closest proximity, both reflecting realistic threat escalation patterns.

**Q2 & Q6:** Delay risk and cargo threat risk exhibit only mild increases near the storm's core (09-16 00:00 and 09-18 00:00), with values close to zero at other times. This reflects limited exposure to direct impact for most routes, which are primarily influenced by peripheral winds—an expected outcome.

**Q3:** The reroute demand score rises sharply to approximately 60% starting from 09-15 12:00, indicating that once spatial impact levels (Q1) exceed a certain threshold, rerouting behavior is activated accordingly. This aligns with practical ripple-effect strategies in maritime logistics.

**Q4:** Duration impact remains relatively stable at 10–11%, primarily driven by the assumed 10% detour increase in our simulation setup. This matches theoretical expectations under moderate rerouting conditions.

**Q5:** Port congestion scores increase to around 20% during high-impact periods (09-15 12:00 and 09-18 00:00), reflecting secondary congestion effects caused by flow disruptions—a network-level amplification effect consistent with congestion theory.

**Q7:** Most of the time, values remain below 2%, indicating minimal gust impact and stable wind fields. This is consistent with computed wind speed gradients along midpoints of typical routes.

**Q8:** As a complementary indicator to Q1, the uncertainty score is high (above 75%) when the typhoon is still distant, and peaks around 55% during periods of greatest forecast ambiguity. This behavior logically follows expected uncertainty patterns in early-stage cyclone evolution.

**Robustness of Prompt-based Warning Vectors.** To further validate the reliability of the LLM-derived warning vectors, we conduct two robustness experiments for the 48-hour prediction horizon on both maritime transportation networks.

**(a) Sensitivity to different binarization thresholds.** Recall that each continuous warning score $[Q_1, \ldots, Q_8] \in [0, 100]^8$ is converted to a binary treatment vector $B_t \in \{0, 1\}^8$ via $b_i^t = \mathbb{I}\{q_i^t > \tau\}$. In Figure 3(a), we vary a global threshold $\tau \in \{0, 10, 20, \ldots, 90\}$ and re-train RippleNet. The case $\tau = 0$ corresponds to an event-only setting in which all non-zero scores are treated as active warnings. As $\tau$ increases from 0 to a moderate range (around 30–60), the MAE decreases and then stabilizes, indicating that filtering out low-confidence scores removes spurious treatments while preserving informative warning signals. Extremely large thresholds ($\tau \geq 70$) lead to a mild degradation, as most warnings are discarded and the model behaves closer to a "no-warning" regime. The broad performance plateau in the mid-range suggests that the LLM-derived warning vectors provide a robust treatment representation.

**(b) Ablation on warning-state dimensions** ($Q_1$–$Q_8$)**.** We also quantify the contribution of each warning dimension. Starting from the full 8-dimensional warning vector, we construct eight *leave-one-dimension-out* variants: $-Q_k$ removes the $k$-th dimension before binarization, while keeping all other components and training settings unchanged. Figure 3(b) reports the 5-run average MAE (mean $\pm$ std) for the full model and all $-Q_k$ variants. Removing any single dimension leads to a noticeable degradation, confirming that all eight questions carry useful causal information. The largest MAE increases occur when $Q_1$ (spatial impact), $Q_3$ (reroute need), or $Q_5$ (port congestion) are removed, especially in the denser East Asia network, which matches operational intuition about the importance of these factors in warning-induced ripple effects. In contrast, dropping $Q_2$ (delay risk), $Q_6$ (speed adjustment), $Q_7$ (cargo threat), or $Q_8$ (uncertainty level) yields smaller but still consistent performance losses, indicating that they play complementary roles rather than being redundant.

