# OpenReview forum: "RippleNet: Learning Causal Maritime Dynamics for Forecasting Warning-Induced Ripple Effects"
_ICLR.cc/2026/Conference — Submitted to ICLR 2026_

### Official Review · Reviewer_ijxk · 2025-10-20

**Soundness:** 2
**Presentation:** 3
**Contribution:** 1
**Rating:** 2
**Confidence:** 4

**Summary:**

The paper proposes RippleNet, a causal spatio-temporal framework for forecasting warning-induced ripple effects in maritime networks. It combines a neural deconfounder, an ODE-based propagation module, and LLM-generated warning vectors. Experiments on real-world datasets show improved forecasting performance over existing baselines.

**Strengths:**

1. This work has strong potential impact in the domain of maritime logistics and global trade. The proposed framework, if validated and deployed, could significantly improve the understanding and management of warning-induced disruptions in international shipping routes.

2. The figures are clear and well-organized, and the inclusion of map-based visualizations greatly enhances interpretability.

**Weaknesses:**

1. The reviewer’s primary concern is that the paper’s contributions to the representation learning and machine learning community appear limited. The work emphasizes domain-specific applications (maritime forecasting) rather than advancing general ML methodology. It may be more suitable for a journal, conference, or workshop focused on maritime operations or applied geospatial modeling.

2. The paper lacks any discussion or quantitative analysis regarding computational efficiency, including model size, parameter count, training speed, and GPU memory usage. As the proposed model involves an ODE-based causal propagation module, its computational overhead relative to other baseline models should be reported for a fair assessment.

3. The hyperparameter settings of baseline models (e.g., STGCN, GraphWaveNet, PDFormer, CaST) are not described. It is unclear whether they were tuned on validation sets or directly adopted from the respective papers, which limits the reproducibility of the reported comparisons.

4. The paper’s reliance on LLM-generated warning vectors raises concerns about reproducibility, transparency, and robustness. While Appendix A.2 provides detailed prompt templates and a rationality analysis, the use of a proprietary LLM (ChatGPT-4o) introduces potential subjectivity and temporal variability in outputs. The validation is primarily qualitative; no quantitative reliability assessment (e.g., inter-model consistency or expert annotation agreement) is provided. Without rigorous verification, there remains a risk that hallucinations or latent biases in the LLM outputs could propagate into the causal modeling pipeline, undermining its interpretive validity.

5. The sensitivity and ablation analyses are limited in scope. Figure 5 only examines two hyperparameters (λ and α). A more comprehensive robustness analysis—considering solver tolerances, LLM threshold values, and state vector dimensionality—would strengthen the empirical section. Similarly, the ablation study isolates only top-level modules, leaving finer-grained dependencies (e.g., within the LLM vector representation) unexplored.

6. Theoretical validation of the causal adjustment component remains underdeveloped. Although the equations are sound and conceptually aligned with back-door adjustment, there is no formal discussion of identifiability, convergence, or conditions under which the deconfounder reliably works. Such analysis would enhance the methodological rigor of the paper.

**Questions:**

1. Could the proposed framework be extended or tested on additional regions, such as American or African maritime networks, to demonstrate cross-regional generalizability?

2. Is RippleNet applicable beyond the maritime transportation domain, for example, to air traffic, railway, or supply chain networks affected by external disruptions?

3. How sensitive is the model’s performance to the quality and consistency of the LLM-generated warning vectors? Have the authors compared outputs across different LLMs (e.g., Claude, Gemini) or evaluated agreement with expert annotations?

4. Could the authors provide quantitative details on model efficiency, such as parameter count, runtime, or GPU memory consumption, to contextualize the cost-performance tradeoff against baseline models?

5. Have the authors considered introducing formal causal validation metrics, such as Average Causal Effect estimation or counterfactual consistency tests, to strengthen the causal claims?

6. Would a more extensive hyperparameter and solver sensitivity analysis, for example, varying the ODE tolerance or hidden dimensionality, alter the observed performance trends?

---

> ### Author Response · Authors · 2025-11-21
> **Response to Reviewer ijxk (Q1-Q2)**
>
> Thank you for your review. We respectfully disagree that our contributions are purely domain-specific.
>
> **1. General causal spatio-temporal framework.**
> RippleNet is designed as a generic causal spatio-temporal architecture for forecasting on dynamic graphs under interventions, not a bespoke maritime model. Its three core components—(i) a neural backdoor-style deconfounder over $(X, V, E)$, (ii) a continuous-time ODE propagation module for interference on graphs, and (iii) a structured treatment vector $B$ derived from unstructured warnings—are defined at the level of abstract graph time series and can be applied to other domains with networked flows and exogenous interventions.
>
> **2. Methodological advances beyond existing STGNN and causal baselines.**
> We go beyond “applying an existing model to a new dataset” by (i) introducing a treatment-aware deconfounder block that operationalizes backdoor adjustment inside a deep ST model, and (ii) modeling warning-induced ripple effects via a learnable ODE with explicit spatial decay and propagation, instead of purely correlational attention or message passing. These are generic building blocks for learning causal representations and dynamics on graphs.
>
> **3. Benchmarking causal forecasting under warnings.**
> We also introduce a realistic setting for causal forecasting on large graphs under structured treatments, directly targeting robustness under interventions and OOD conditions—issues of broad interest to the ML community.
>
> We then address each concern below.
>
> ---
>
> $Q1.$ **Could the proposed framework be extended or tested on additional regions**
>
> Conceptually, RippleNet is not tied to East Asia or Northwest Europe: the framework only assumes (i) a dynamic flow graph, (ii) structured warning/treatment vectors, and (iii) pre-treatment contextual covariates \((X, V, E)\). Given AIS trajectories, port metadata, and warning sources, the same pipeline could be instantiated for American or African maritime networks without changing the core architecture. In practice, however, we do not yet have processed datasets for these regions, so instead we **have added a cross-region transfer experiment** between two distinct maritime networks (Table 5 in Appendix A.2). RippleNet consistently outperforms CaST, a causal graph–based baseline, in both transfer directions between East Asia and Northwest Europe (e.g., MAE 1.427 vs. 1.528 and 2.649 vs. 2.796), indicating that the learned representation generalizes well across regions and yields more stable treatment–outcome relationships.
>
> **Table 5.**  Transfer learning performance of CaST and RippleNet between the East Asia and Northwest Europe maritime transportation networks (5-run mean ± std).
>
> | Source Dataset | Target Dataset    | Metric | CaST              | RippleNet           |
> | :-------- | :---------------- | :----: | :---------------- | :------------------ |
> | **East Asia** | **Northwest Europe** | MAE    | 1.528 ± 0.254 | **1.427 ± 0.143** |
> |           |                   | RMSE   | 3.350 ± 0.592 | **3.169 ± 0.353** |
> | **Northwest Europe** | **East Asia** | MAE    | 2.796 ± 0.248 | **2.649 ± 0.191** |
> |           |                   | RMSE   | 6.854 ± 0.480 | **6.518 ± 0.416** |
>
> ---
>
> $Q2.$ **Beyond the maritime transportation domain, for example, to air traffic, railway, or supply chain networks affected by external disruptions**
>
> Yes – conceptually:
>
> * **Rail networks:** nodes = stations; edges = rail lines; warnings = maintenance/weather/accidents.
> * **Air traffic:** nodes = airports; edges = routes; warnings = weather/airspace closures.
> * **Supply chains:** nodes = warehouses/factories; edges = transport links; warnings = disasters, strikes, policy shocks.
>
> In all these cases, we have:
>
> * A network with flow-like quantities,
> * Disruption events with textual descriptions, and
> * Cascading effects naturally modeled via ODE-like dynamics.
>
> Our architecture was designed with this generality in mind. The key requirements are:
>
> 1. **Network structure:** spatial graph with measurable flows.
> 2. **Event descriptions:** textual warnings that can be quantified by an LLM.
> 3. **Historical data:** time series of flows and events for training.
>
> For example, in **rail networks**, we could:
>
> * Treat stations as nodes and rail lines as edges;
> * Use passenger counts or freight volumes as flows;
> * Feed maintenance notices, weather warnings, or accident reports into the LLM warning encoder;
> * Learn how these events propagate using the same ODE-based dynamics.
>
> Similarly, for **supply chains**, disruptions like port strikes, factory closures, or policy changes could be encoded as warnings, and RippleNet could predict how shipment flows adjust across the logistics network. We will emphasize such extensions in the conclusion and discuss domain-specific adaptations.

---

> ### Author Response · Authors · 2025-11-21
> **Response to Reviewer ijxk (Q3-Q5)**
>
> $Q3.$ **Sensitivity to LLM-generated warning vectors and comparison across models/experts**
>
> Thank you for this valuable comment. We acknowledge that our previous hyperparameter sensitivity analysis for the LLM-derived warning-vector treatment was limited, so we have added **new robustness experiments for LLM-derived warning vectors** in Appendix A.3. Specifically, for the 48-hour prediction horizon on both maritime transportation networks, we evaluate (i) sensitivity to different binarization thresholds and (ii) ablations over warning-state dimensions ($Q_1$–$Q_8$) to further validate the reliability of the prompt-based warning vectors.
>
> Regarding multiple LLMs, we deliberately fixed a **single** off-the-shelf model (GPT-4o) with deterministic decoding settings (temperature = 0, top-p = 1.0) to (i) avoid introducing an additional LLM-selection axis into the study, and (ii) ensure that warning-vector generation is reproducible given the same prompt and inputs.
>
> ---
>
> $Q4.$ **Provide the computational cost**
>
> In response, we have added a computational efficiency analysis in Appendix A.1. Specifically, Table 2 (in Appendix) reports the average wall-clock training time per epoch and peak GPU memory usage for all baselines and RippleNet on both maritime transportation networks. Classical GNN-based models (e.g., STGCN, GraphWaveNet) are the fastest and most lightweight, while transformer-based architectures (PDFormer, STAEformer, CaST) incur noticeably higher costs. RippleNet lies in a moderate regime: it is slower and more memory-consuming than the simplest GNNs due to the causal deconfounder and ODE modules, but remains substantially more efficient than STAEformer and comparable to other transformer-style models, with per-epoch training time below 14 seconds and peak memory usage under 8 GB on East Asia.
>
> **Table 2. Training speed and GPU memory usage among baselines and our proposed method. Here, efficiency values are reported as East Asia / Northwest Europe.**
>
> | Metric \ Model        | STGCN        | MTGNN        | AGCRN        | GMAN         | GraphWaveNet | PDFormer      | STAEformer    | CaST         | RippleNet    |
> |-----------------------|--------------|--------------|--------------|--------------|--------------|--------------|--------------|--------------|--------------|
> | Time per epoch (s)    |  0.84 / 0.59      | 1.82 / 1.62      | 3.36 / 3.18      |  5.39 / 2.49      | 3.51 / 1.64      | 8.36 / 3.80      |  246.61 / 115.23       | 5.97 / 2.83      | 13.85 / 7.79       |
> | GPU memory (GB)  | 0.11 / 0.07      | 0.32 / 0.19      | 2.17 / 1.21      | 2.83 / 1.68      | 1.58 / 0.94      | 4.17 / 2.50      |  15.51 / 9.08       |   3.56 / 2.39    | 7.15 /  4.74   |
>
> ---
>
> $Q5.$ **Introducing formal causal validation metrics**
>
> We agree that formal causal validation metrics are important in principle, and we have considered using quantities such as the Average Causal Effect (ACE). However, the “true” ACE cannot be directly evaluated from our real-world data, since each route is only observed under its factual warning history and we lack randomized interventions or ground-truth counterfactual outcomes. While our neural deconfounder does not learn explicit propensity scores, we provide several empirical checks:
>
> 1. **Ablation evidence (Table 3 in main paper).** Removing the neural back-door adjustment (RippleNet-DEF) raises MAE by 10.7%/18.7%, suggesting that causal disentanglement is key to mitigating confounding under warning events.
>
> 2. **Interpretability (Figure 7 in main paper).** We compute effect magnitudes by intervening on the warning treatment within the model (setting $B=0$ vs. $B=1$) and visualizing $|Y_1 - Y_0|$ across space and time. While this is not a ground-truth ACE, the resulting patterns align with domain knowledge about typhoon impacts (strong suppression on exposed routes, weaker effects on peripheral links), providing a rationality check for the learned causal effects.
>
> 3. **Cross-region transfer (Table 5 in Appendix A.2).** RippleNet consistently outperforms CaST, a causal graph–based baseline, in both transfer directions between East Asia and Northwest Europe (e.g., MAE 1.427 vs. 1.528 and 2.649 vs. 2.796), indicating that the learned representation generalizes well across regions and yields more stable treatment–outcome relationships.

---

> ### Author Response · Authors · 2025-11-21
> **Response to Reviewer ijxk (Q6)**
>
> $Q6.$ **More extensive hyperparameter (e.g., LLM threshold values and state vector dimensionality) and solver sensitivity analysis**
>
> Regarding extensive hyperparameter analysis, we acknowledge that our previous hyperparameter sensitivity analysis for the LLM-derived warning-vector treatment was limited. So we have **added a more detailed hyperparameter analysis**: we compare RippleNet with CFR, TarNet, CaST, and two treatment variants (RippleNet-CT and RippleNet-Event). CFR and TarNet learn balanced latent spaces without graphs, while CaST is a causal GNN with separate spatial/temporal paths. In RippleNet, treatment is an 8-dimensional **binary** warning vector with **dimension-specific thresholds** from maritime domain knowledge; RippleNet-CT instead uses continuous LLM warning scores, and RippleNet-Event collapses all warnings into a single binary flag. Table 4 (in Appendix) shows that all **causal + treatment-aware** models outperform purely correlational GNNs, with RippleNet achieving the best performance: its thresholded 8-dimensional binary treatment yields a clean, intervention-ready representation that filters noisy low-confidence scores while preserving heterogeneous warning dimensions, whereas RippleNet-CT is more sensitive to score calibration and RippleNet-Event discards most structured warning semantics.
>
> **Table 4.** Performance comparison of causal baselines and different treatment variants of our method on both maritime transportation networks.
>
> | Dataset | Metric | CFR | TarNet | CaST | RippleNet | RippleNet-CT | RippleNet-Event |
> | :--- | :---: | :---: | :---: | :---: | :---: | :---: | :---: |
> | **East Asia** | MAE  | 2.654 | 2.727 | 2.404 | **2.162** | 2.174 | 2.278 |
> |               | RMSE | 6.630 | 6.845 | 6.368 | **6.186** | 6.203 | 6.229 |
> | **Northwest Europe** | MAE  | 1.476 | 1.502 | 1.392 | **1.175** | 1.178 | 1.194 |
> |                     | RMSE | 2.991 | 3.152 | 3.053 | **2.798** | 2.805 | 2.861 |
>
> Regarding ODE solver and theoretical validation of the causal adjustment, we **have reorganized** the *Methodology* and added a dedicated subsection in Appendix A.1 (Causal Semantics and Deconfounder Implementation) that makes our causal assumptions explicit:
>
> 1. We clearly distinguish (i) raw warning text $A$, (ii) the structured treatment vector $B$, and (iii) pre-treatment covariates $(\mathbf{X}, \mathbf{V}, \mathbf{E})$, and formally state that our estimand is $P(\mathbf{Y} \mid \text{do}(B))$.
>
> 2. We explain how the neural deconfounder block implements a back-door–style adjustment in representation space, using interaction gates $(\mathbf{C}{BX}, \mathbf{C}{BV}, \mathbf{C}_{BE})$ to down-weight associations between $B$ and $\mathbf{Y}$ that are explained by pre-treatment context, while preserving residual variation attributable to genuine warning effects.
>
> 3. We also describe the ODE-based ripple propagation more concretely: it is instantiated with a Dormand–Prince RK45 solver (via torchdiffeq) and interpreted as an explicit interference model, where changes in $B$ alter local causal representations $\mathbf{Z}_{\text{causal}}$ and then propagate along graph edges with spatial decay and dual-exponential temporal kernels.
>
> In addition, we fix the ODE solver tolerances to stable default values across all experiments. Preliminary tests with tighter and looser tolerances showed performance changes well within the run-to-run variance, so we treat these tolerances as an implementation detail rather than a primary hyperparameter.
>
> ---
>
> **Additional clarification: hyperparameter settings of baseline models**
>
> We apologize for the missing clarification. For all baselines, we start from the official implementations and hyperparameter configurations recommended in the original papers, and then perform tuning on our validation sets rather than directly adopting them unchanged. We have added this point, together with the exact train/validation/test splits, to Appendix A.1 to improve reproducibility.

---

> ### Comment · Reviewer_ijxk · 2025-11-28
>
> The reviewer sincerely thanks the authors for their comprehensive and carefully prepared response, which has helped resolve many of the previous concerns. While many concerns have been effectively addressed, some aspects may still require further clarification.
>
> The reviewer acknowledges the authors’ intent to propose a general framework. However, to substantiate this claim, it would be expected to demonstrate superiority over established baseline methods such as GraphWaveNet, PDFormer, STAEformer, and CaST using widely adopted benchmark datasets that are commonly employed in prior studies. At present, such evidence appears to be limited. The proposed approach seems to be evaluated primarily under a specific scenario, with a focus on the maritime and shipping logistics domains. Moreover, all experimental datasets, such as those from Northwest Europe and East Asia, pertain to this domain. This may suggest that the work is more application-oriented rather than a generalizable machine learning or representation learning methodology aligned with the scope of ICLR. This observation appears to be in contrast with the authors’ claim of presenting a general framework.
>
> The commonly used benchmark datasets in baseline models are as follows:
>
> - PEMS04 – used in PDFormer and STAEformer
> - PEMS07 – used in PDFormer and STAEformer
> - PEMS08 – used in CaST and PDFormer
> - METR-LA – used in GraphWaveNet and STAEformer
> - PEMS-BAY – used in GraphWaveNet and STAEformer
> - AIR-BJ – used in CaST
> - AIR-GZ – used in CaST
>
> Could the authors conduct additional experiments using these datasets? Doing so would further reinforce the authors’ claim that the proposed method serves as a general framework.
>
> The reviewer is still carefully considering whether to adjust the score. While some concerns have been partially alleviated through the authors’ additional experiments and comprehensive responses, the most critical concern remains unaddressed. The reviewer looks forward to further clarification from the authors.
>
> Sincerely,
>
> Reviewer ijxk

---

> > ### Author Response · Authors · 2025-12-01
> >
> > We thank the reviewer for pointing out the concern about generality of our proposed method. We fully agree that, to support the claim of a general framework, it is important to demonstrate that **RippleNet** performs well **beyond maritime logistics**.
> >
> > ---
> >
> > ### 1. New experiments on the AIR-BJ benchmark
> >
> > Following the reviewer’s suggestion, we have added new experiments on the **AIR-BJ** air quality dataset. We use the KDD Cup 2018 PM2.5 dataset for Beijing, spanning **2017-01-01 to 2017-12-31** with hourly resolution, covering 35 monitoring stations. For the warning variables, we construct a weather-based warning dataset from **ERA5** over the Beijing region and extract **temperature, wind, and precipitation** for corresponding historical timestamps. These weather features are then fed into our prompt-based generation process to derive warning vectors conditioned on specific weather thresholds. The data are split into training/validation/test sets with a ratio of **4:1:1**.
> >
> >
> > Under this setting, RippleNet is compared against STGCN, MTGNN, AGCRN, GMAN, GraphWaveNet, PDFormer, STAEformer, and CaST. The experimental results are shown below.
> >
> > **Table 1.** Performance comparison of different graph-based models for air quality prediction (**AIR-BJ**). The best results are marked in bold.
> >
> > | Horizon | Metric | STGCN | MTGNN | AGCRN | GMAN | GraphWaveNet | PDFormer | STAEformer | CaST | RippleNet |
> > |--------|--------|-------|-------|-------|------|--------------|----------|------------|------|-----------|
> > | 6 h    | MAE    | 21.78 | 23.04 | 23.27 | 20.38 | 19.87 | 19.08 | 19.57 | 18.86 | **18.17** |
> > | 6 h    | RMSE   | 38.31 | 40.20 | 40.49 | 36.44 | 35.76 | 34.91 | 35.32 | 33.40 | **32.74** |
> > |--------|--------|-------|-------|-------|------|--------------|----------|------------|------|-----------|
> > | 12 h   | MAE    | 28.56 | 30.42 | 30.94 | 27.35 | 26.52 | 25.84 | 25.73 | 24.39 | **23.26** |
> > | 12 h   | RMSE   | 48.15 | 50.29 | 50.83 | 47.26 | 46.65 | 46.22 | 46.69 | 45.41 | **44.93** |
> >
> > RippleNet **consistently outperforms all baselines** on a **non-maritime, urban air-quality** forecasting task. This provides direct evidence that the proposed framework is not restricted to maritime logistics.
> >
> > ---
> >
> > ### 2. Why AIR-GZ and PeMS-style datasets are not included
> >
> > We attempted to include **AIR-GZ** for completeness. However, to the best of our knowledge and efforts, the version of AIR-GZ datasets used by CaST is **not publicly available**.
> >
> > We also appreciate the suggestion to evaluate on **PEMS04/07/08, METR-LA, and PEMS-BAY**. However, these benchmarks are not well aligned with the core setting that RippleNet is designed to address, for two main reasons:
> >
> > 1. **Short temporal coverage and limited disruptions.**
> >    Most PeMS-style datasets span only about **2–5 months** and mainly capture regular commuting patterns. RippleNet targets **warning-induced ripple effects** that accumulate over longer periods and across diverse extreme events. Short time series with few major disruptions provide limited opportunity to meaningfully evaluate these causal ripple mechanisms.
> >
> > 2. **No explicit warning/intervention signals.**
> >    RippleNet relies on **explicit treatment variables (warning vectors)**, derived from domain warnings and used for back-door adjustment and continuous-time ripple propagation. Standard PeMS datasets provide traffic measurements (and sometimes basic weather), but **no explicit annotations for incidents, control measures, or system-level alerts**.

---

### Official Review · Reviewer_SRqw · 2025-11-01

**Soundness:** 3
**Presentation:** 3
**Contribution:** 3
**Rating:** 6
**Confidence:** 4

**Summary:**

This paper introduces RippleNet, a causal spatio-temporal forecasting framework designed to model and predict warning-induced disruptions (“ripple effects”) in maritime transportation networks. The authors identify a fundamental limitation of existing traffic-forecasting methods — their dependence on correlations rather than causal mechanisms — which leads to failure under anomalous warning scenarios (e.g., typhoons, security alerts).

RippleNet integrates three key components:
1. Neural Deconfounder Block — performs causal adjustment via neural back-door control to disentangle genuine effects of warnings from spurious correlations.
2. Continuous-Time ODE Propagation Module — models how disruptions spread temporally and spatially across port networks.
3. LLM-Generated Warning Vectors — quantifies multidimensional warning impacts from textual maritime bulletins using large-language-model prompts.

Experiments on East Asia and Northwest Europe maritime flow datasets demonstrate consistent gains (up to 19.6% MAE improvement over strong baselines such as PDFormer and STAEformer). Ablation and case studies support the contribution of each module and show interpretable causal propagation patterns.

**Strengths:**

- Originality: Innovative combination of causal inference, ODE modeling, and LLM-based treatment quantification; new causal formulation tailored for maritime ripple-effect prediction.


- Quality: Strong empirical validation with diverse baselines, ablations, and case studies; interpretable visualizations of causal effects (Fig. 7).


- Clarity: Comprehensive methodological exposition, including formal causal graph and mathematical derivations.


- Significance: Provides a blueprint for applying causal deep learning to real-world network resilience problems beyond transportation (e.g., logistics, climate-impact forecasting).


- Reproducibility: Code and datasets are described in detail, supporting transparency.

**Weaknesses:**

- Causal Validity of LLM Inputs: The reliance on LLM-generated binary treatment vectors introduces potential noise and bias; authors should analyze sensitivity to prompting or threshold choices.


- Generalization Beyond Maritime Domain: While experiments are strong, additional non-maritime datasets (e.g., air-traffic or logistics networks) could demonstrate broader applicability.


- Complexity vs. Interpretability: RippleNet’s multi-module architecture may challenge operational deployment; authors could discuss computational cost and parameter efficiency.


- Ablation Depth: The ablation focuses on module removal; further analysis on causal disentanglement metrics or counterfactual validation would reinforce claims.


- Minor Clarity Issues: Dense notation (especially in §4) and lengthy references may obscure key insights for readers unfamiliar with causal ODEs.

- Typographic Consistency: The manuscript occasionally uses incorrect opening quotation marks (e.g., ”ripple effect”, ”Nanmadol”), where the closing quote glyph is mistakenly used at the start. While minor, this distracts from an otherwise polished presentation and should be corrected.

**Questions:**

1. How sensitive is performance to the binarization thresholds (τᵢ) and the choice of LLM prompt design?


2. Could the authors provide quantitative evidence that the learned deconfounder truly captures causal rather than correlational signals (e.g., intervention or counterfactual tests)?


3. What are the computational costs of the ODE solver, and how does it scale with network size?


4. Could RippleNet be adapted for continuous (non-binary) treatment variables or multi-modal warning inputs (e.g., satellite imagery, textual logs)?


5. How do the authors plan to address domain shift when applying to unseen maritime regions or future warning types?

---

> ### Author Response · Authors · 2025-11-21
> **Response to Reviewer SRqw (Q1-Q3)**
>
> Thank you for the positive assessment and for recognizing the novelty of integrating causal inference with neural ODEs. We address your questions below.
>
> ---
>
> $Q1.$ **How sensitive is performance to the binarization thresholds and the choice of LLM prompt design**
>
> Thank you for this valuable comment. We acknowledge that our previous hyperparameter sensitivity analysis for the LLM-derived warning-vector treatment was limited, so we have **added new robustness experiments** in Appendix A.3. Specifically, for the 48-hour prediction horizon on both maritime transportation networks, we evaluate (i) sensitivity to different binarization thresholds and (ii) ablations over warning-state dimensions ($Q_1$–$Q_8$) to further validate the reliability of the prompt-based warning vectors.
>
> ---
>
> $Q2.$ **Provide quantitative evidence that the learned deconfounder truly captures causal rather than correlational signals (e.g., intervention or counterfactual tests)**
>
> While our neural deconfounder does not learn explicit propensity scores, we provide several empirical checks, including a newly added experiment on cross-region transfer between different maritime networks:
>
> 1. **Ablation evidence (Table 3 in main paper).** Removing the neural back-door adjustment (RippleNet-DEF) raises MAE by 10.7%/18.7%, confirming that causal disentanglement is key to mitigating confounding under warning events.
>
> 2. **Interpretability (Figure 7 in main paper).** The learned warning effects align with maritime domain knowledge, with larger effects of typhoon on major exposed routes and smaller effects on peripheral links, which is more consistent with a causal than a purely correlational signal.
>
> 3. **Could be used for region transfer (Table 5 in Appendix A.2).** RippleNet consistently outperforms CaST, a causal graph–based baseline, in both transfer directions between East Asia and Northwest Europe (e.g., MAE 1.427 vs. 1.528 and 2.649 vs. 2.796), indicating that the learned representation generalizes well across regions and yields more stable treatment–outcome relationships.
>
> **Table 5.**  Transfer learning performance of CaST and RippleNet between the East Asia and Northwest Europe maritime transportation networks (5-run mean ± std).
>
>
> | Source Dataset | Target Dataset    | Metric | CaST              | RippleNet           |
> | :-------- | :---------------- | :----: | :---------------- | :------------------ |
> | **East Asia** | **Northwest Europe** | MAE    | 1.528 ± 0.254 | **1.427 ± 0.143** |
> |           |                   | RMSE   | 3.350 ± 0.592 | **3.169 ± 0.353** |
> | **Northwest Europe** | **East Asia** | MAE    | 2.796 ± 0.248 | **2.649 ± 0.191** |
> |           |                   | RMSE   | 6.854 ± 0.480 | **6.518 ± 0.416** |
>
> Taken together, this evidence is empirically consistent with improved deconfounding in the learned representation.
>
> ---
>
> $Q3.$ **What are the computational costs of the ODE solver, and how does it scale with network size?**
>
> In response, we have added a computational efficiency analysis in Appendix A.1. Specifically, Table 2 (in Appendix) reports the average wall-clock training time per epoch and peak GPU memory usage for all baselines and RippleNet on both maritime transportation networks. Classical GNN-based models (e.g., STGCN, GraphWaveNet) are the fastest and most lightweight, while transformer-based architectures (PDFormer, STAEformer, CaST) incur noticeably higher costs. RippleNet lies in a moderate regime: it is slower and more memory-consuming than the simplest GNNs due to the causal deconfounder and ODE modules, but remains substantially more efficient than STAEformer and comparable to other transformer-style models, with per-epoch training time below 14 seconds and peak memory usage under 8 GB on East Asia.
>
> **Table 2. Training speed and GPU memory usage among baselines and our proposed method. Here, efficiency values are reported as East Asia / Northwest Europe.**
>
> | Metric \ Model        | STGCN        | MTGNN        | AGCRN        | GMAN         | GraphWaveNet | PDFormer      | STAEformer    | CaST         | RippleNet    |
> |-----------------------|--------------|--------------|--------------|--------------|--------------|--------------|--------------|--------------|--------------|
> | Time per epoch (s)    |  0.84 / 0.59      | 1.82 / 1.62      | 3.36 / 3.18      |  5.39 / 2.49      | 3.51 / 1.64      | 8.36 / 3.80      |  246.61 / 115.23       | 5.97 / 2.83      | 13.85 / 7.79       |
> | GPU memory (GB)  | 0.11 / 0.07      | 0.32 / 0.19      | 2.17 / 1.21      | 2.83 / 1.68      | 1.58 / 0.94      | 4.17 / 2.50      |  15.51 / 9.08       |   3.56 / 2.39    | 7.15 /  4.74   |

---

> > ### Comment · Reviewer_SRqw · 2025-11-25
> >
> > I appreciate the authors' rebuttal addressing most of my concerns. However, empirical checks are not sufficient to argue that the learned deconfounder truly captures causal rather than correlational signals. Also, I agree with Reviewer ijxk, that the methodological contributions of the paper are limited. I will keep my initial score. Thanks.

---

> > > ### Author Response · Authors · 2025-12-01
> > >
> > > Regarding the reviewer’s concerns about the limitations of our proposed framework, we fully agree that, to substantiate the claim of a general framework, it is important to demonstrate that **RippleNet** performs well **beyond maritime logistics**. To this end, we have added **new experiments** on the **AIR-BJ air quality dataset**:
> > >
> > >
> > > We use the KDD Cup 2018 PM2.5 dataset for Beijing, spanning **2017-01-01 to 2017-12-31** with hourly resolution, covering 35 monitoring stations. For the warning variables, we construct a weather-based warning dataset from **ERA5** over the Beijing region and extract **temperature, wind, and precipitation** for corresponding historical timestamps. These weather features are then fed into our prompt-based generation process to derive warning vectors conditioned on specific weather thresholds. The data are split into training/validation/test sets with a ratio of **4:1:1**.
> > >
> > >
> > > Under this setting, RippleNet is compared against STGCN, MTGNN, AGCRN, GMAN, GraphWaveNet, PDFormer, STAEformer, and CaST. The experimental results are shown below.
> > >
> > > **Table 1.** Performance comparison of different graph-based models for air quality prediction (**AIR-BJ**). The best results are marked in bold.
> > >
> > > | Horizon | Metric | STGCN | MTGNN | AGCRN | GMAN | GraphWaveNet | PDFormer | STAEformer | CaST | RippleNet |
> > > |--------|--------|-------|-------|-------|------|--------------|----------|------------|------|-----------|
> > > | 6 h    | MAE    | 21.78 | 23.04 | 23.27 | 20.38 | 19.87 | 19.08 | 19.57 | 18.86 | **18.17** |
> > > | 6 h    | RMSE   | 38.31 | 40.20 | 40.49 | 36.44 | 35.76 | 34.91 | 35.32 | 33.40 | **32.74** |
> > > |--------|--------|-------|-------|-------|------|--------------|----------|------------|------|-----------|
> > > | 12 h   | MAE    | 28.56 | 30.42 | 30.94 | 27.35 | 26.52 | 25.84 | 25.73 | 24.39 | **23.26** |
> > > | 12 h   | RMSE   | 48.15 | 50.29 | 50.83 | 47.26 | 46.65 | 46.22 | 46.69 | 45.41 | **44.93** |
> > >
> > > RippleNet **consistently outperforms all baselines** on a **non-maritime, urban air-quality** forecasting task. This provides direct evidence that the proposed framework is not restricted to maritime logistics.

---

> ### Author Response · Authors · 2025-11-21
> **Response to Reviewer SRqw (Q4-Q5)**
>
> $Q4.$ **Ablation depth & continuous (non-binary) treatment variables**
>
> In response, we have newly added corresponding experiments in the Appendix, including causal baselines and treatment variants:
>
> To better isolate the effect of explicit treatment modeling, we compare RippleNet with CFR, TarNet, CaST, and two treatment variants (RippleNet-CT and RippleNet-Event). CFR and TarNet learn balanced latent spaces without graphs, while CaST is a causal GNN with separate spatial/temporal paths. In RippleNet, treatment is an 8-dimensional **binary** warning vector with **dimension-specific thresholds** from maritime domain knowledge; RippleNet-CT instead uses continuous LLM warning scores, and RippleNet-Event collapses all warnings into a single binary flag. Table 4 (in Appendix) shows that all **causal + treatment-aware** models outperform purely correlational GNNs, with RippleNet achieving the best performance: its thresholded 8-dimensional binary treatment yields a clean, intervention-ready representation that filters noisy low-confidence scores while preserving heterogeneous warning dimensions, whereas RippleNet-CT is more sensitive to score calibration and RippleNet-Event discards most structured warning semantics.
>
> **Table 4.** Performance comparison of causal baselines and different treatment variants of our method on both maritime transportation networks.
>
> | Dataset | Metric | CFR | TarNet | CaST | RippleNet | RippleNet-CT | RippleNet-Event |
> | :--- | :---: | :---: | :---: | :---: | :---: | :---: | :---: |
> | **East Asia** | MAE  | 2.654 | 2.727 | 2.404 | **2.162** | 2.174 | 2.278 |
> |               | RMSE | 6.630 | 6.845 | 6.368 | **6.186** | 6.203 | 6.229 |
> | **Northwest Europe** | MAE  | 1.476 | 1.502 | 1.392 | **1.175** | 1.178 | 1.194 |
> |                     | RMSE | 2.991 | 3.152 | 3.053 | **2.798** | 2.805 | 2.861 |
>
> In addition, for multi-modal warning inputs, our framework only assumes a structured treatment vector \(B\), so we could replace the current text-based encoder with a multi-modal encoder (e.g., combining satellite imagery and textual logs) to produce \(B\), while keeping the neural deconfounder and ODE propagation modules unchanged. A full multi-modal design is an interesting direction for future work.
>
> ---
>
> $Q5.$ **Address domain shift when applying to unseen maritime regions or future warning types**
>
> Conceptually, RippleNet separates (i) generic spatio-temporal dynamics (captured by the neural deconfounder and ODE propagation) from (ii) the warning representation \(B\). When moving to a new region or new warning types, one can keep the core RippleNet architecture and adapt only the encoder that maps raw warning information to \(B\) (e.g., via re-prompting or light fine-tuning of the LLM mapping). To validate this point, we have included a newly added experiment through cross-region experiments between East Asia and Northwest Europe (Appendix A.2, Table 5), where RippleNet consistently outperforms CaST even when trained on one region and evaluated on the other, indicating improved robustness to regional differences.
>
> ---
>
> **Additional clarification: clarity issues \& typographic consistency**
>
> We thank the reviewer for these helpful presentation comments.
>
> **Dense notation and long references (§4).** In the revised version, we have added clearer explanatory text in §4 to provide more intuition for the causal ODE formulation. We have also moved additional technical details into Appendix A.1 as a dedicated subsection on *Causal Semantics and Deconfounder Implementation*.
>
> **Typographic consistency (quotation marks).** We have carefully checked the manuscript for incorrect opening quotation marks and corrected all such instances (e.g., “ripple effect”, “Nanmadol”).

---

> ### Author Response · Authors · 2025-11-26
>
> We appreciate your follow-up comments, but we are **frankly confused** by the requirements for counterfactual tests. In principle, intervention or counterfactual tests would indeed be the most direct way to verify whether a learned deconfounder captures causal rather than purely correlational signals. However, in our maritime warning setting, such tests are **intrinsically infeasible in practice**.
>
> First, as we discuss in our motivation (Fig. 1), a typhoon induces **ripple effects far beyond the wind-field area**: traffic on routes that never enter the typhoon area still changes significantly due to rerouting, port closures, and upstream/downstream congestion. As a result, there is no clean way to define, for each individual route and each warning type, a well-delimited “warning period” that only reflects the effect of that specific warning.
>
> Second, different warning types **sometimes overlap in time and space**, and traffic is highly stochastic due to schedule changes, port operations, and commercial decisions. In many cases, it is impossible to attribute the observed flow change on a given route–time pair to one specific warning in a way that would be acceptable as ground truth for an intervention or counterfactual test.
>
> **Similar difficulties** have been documented in other **real-world causal inference tasks** [1, 2]. Ma et al. [1] explicitly argue that randomized controlled trials for COVID-19 policy evaluation are “not readily applicable” and that counterfactual outcomes are unobservable in practice, so causal assessment must instead rely on observational data, proxy variables for unobserved confounders, and **prediction-based sanity checks rather than true experimental interventions**. Zhang et al. [2] further show that typhoon-related impacts propagate widely in space and time in real-world mobility systems, making it difficult to **isolate neatly separated treated and untreated regions or periods**.
>
> Our work follows the same line: we only have observational AIS data and heterogeneous warning bulletins, without the possibility of performing physical interventions or observing ground-truth counterfactual flows at the level of individual routes and warning types. **This is precisely why we formulate the problem as learning a neural deconfounder from rich spatio-temporal proxies (network structure, historical flows, and warning texts) and then validate it through ablations, cross-region transfer, and domain-consistent explainable effect patterns, rather than through idealized intervention/counterfactual tests that are not feasible in this application.** We fully acknowledge that, under these constraints, our empirical evaluation cannot match the strength of evidence provided by randomized interventions; however, **we believe our proposed method offers a rigorous and practically achievable assessment for this type of maritime warning scenario**.
>
>
> [1] J. Ma, Y. Dong, Z. Huang, D. Mietchen, and J. Li. “Assessing the Causal Impact of COVID-19 Related Policies on Outbreak Dynamics: A Case Study in the US.” In *Proceedings of The ACM Web Conference 2022 (WWW ’22)*.
>
> [2] Z. Zhang, H. Wang, Z. Fan, R. Shibasaki, and X. Song. “Assessing the Continuous Causal Responses of Typhoon-related Weather on Human Mobility: An Empirical Study in Japan.” In *Proceedings of the 32nd ACM International Conference on Information and Knowledge Management (CIKM ’23)*.

---

### Official Review · Reviewer_tp7t · 2025-11-01

**Soundness:** 3
**Presentation:** 3
**Contribution:** 3
**Rating:** 8
**Confidence:** 4

**Summary:**

The authors propose a novel causal spatio-temporal framework that explicitly models causal dependencies to predict port-to-port flow disruptions under warning-induced ripple effects. The framework leverages knowledge about warnings (e.g., extreme weather or security alerts) to learn from them and lead to better outcomes. Through several experiments, the authors show that the framework effectively learns causal relationships and that such causal understanding of the context leads to better outcomes than those obtained by correlation-based models. The authors test their framework on two real-world datasets, achieving SOTA performance.

**Strengths:**

The authors propose a novel causal spatio-temporal framework that explicitly models causal dependencies to predict port-to-port flow disruptions under warning-induced ripple effects. They compare their approach against multiple baseline models, surpassing them in performance and achieving SOTA results. They conduct ablation experiments to understand how specific components of the proposed framework contribute to overall performance. Furthermore, they perform hyperparameter analysis and illustrate the usefulness of the proposed framework on a use case. We consider the manuscript to be clearly written and well articulated. The outcomes are likely to have a big impact, as they can directly influence decision-making on logistics worldwide. In addition, the authors created two datasets for assessing the framework. While grounded in publicly available datasets, releasing them would constitute an additional contribution.

**Weaknesses:**

While we consider the work to be solid, we would like to highlight some improvement opportunities:

- The framework requires that warning information be translated into vectors to learn the causal relationships between events and how they translate into outcomes. This translation is done by an LLM, and while a case is described to assess the quality of the outcome, it remains unclear how well it generalizes across many scenarios. Furthermore, it is not clear on what grounds the LLM assigns the scores and whether these are consistent for the same scenario over time, and how robust they are, e.g., to how information is presented to them (a frequent issue as reported by e.g., Leidinger, Alina, Robert Van Rooij, and Ekaterina Shutova. "The language of prompting: What linguistic properties make a prompt successful?." arXiv preprint arXiv:2311.01967 (2023).). No insights were provided on the prompts used to generate these scores or on comparisons across different LLMs.

 - The authors describe a module tasked with controlling for confounding. Nevertheless, little detail is given on how causal relationships are modeled, the embeddings used to model relevant confounders' information, and how they assess the correctness of the information learned.

**Questions:**

1- "This diversity creates challenges for neural networks requiring structured numerical inputs. We address this through large language model-based quantification using domain-specific prompts" -> (i) How reliable are the LLM-generated scores? (ii) In what information are the LLM-generated uncertainty scores grounded? (iii) The authors report average values in Fig. 6a: did they run them several times as to understand consistency and distribution of values? what was the magnitude of the standard deviations? (iv) What kind of models are being used for this purpose? (v) Could this be replaced with an alternative method that, based on the information reported, would provide deterministic estimations for each of the vector values? (vi) Did the authors' study on whether the LLM-generated scores for spatial and duration impact were accurate based on the LLM assessment and the historical data limited to the rationality analysis reported in the Appendix regarding Fig. 6a? (vii) Did the authors cross-check the assessments were accurate ensuring the LLM did not have knowledge about those specific past events (having been trained on such data and therefore creating the illusion of accurate estimation - looking into the future)?, (viii) What prompts did the authors use to extract the information?, (ix) How did the authors test for the robustness of the prompt results to e.g., how the same information reported in different terms (linguistic variations) and order affect the outcomes?, (x) Did the authors consider multiple LLM models and which performed best?

 2- "The threshold values are set [...] based on empirical percentiles and domain knowledge." -> How do the authors ensure proper matching between LLM-generated scores and domain knowledge to arrive at relevant score thresholds?

 3- "and a publicly available maritime warning dataset" -> While the authors provide details regarding the dataset used for East Asia, we could not find details about the one used for Northwest Europe. We encourage the authors to provide a reference to the dataset they used for Northwest Europe.

 4- The authors mention the deconfounding block "controls for confounding by conditioning on the sufficient set of confounding mediators M = {X, V} and spatial-temporal embeddings" and that "the confounding mediators M are modeled through dedicated encoders". We would appreciate it if the authors could provide some insights into (i) how are causal relationships modeled or the authors consider they are implicitly modeled based on the warning vectors and model learning on how these correlate with the outcomes?, (ii) how accurate is to model the confounding mediators with a fixed set of embedding types?, (iii) what kind of embeddings are used in each case?, (iv) are the embeddings modeled using separately trained networks? In such a case, we would appreciate some insights about them.

 5- Table 5: (i) what do the different colour codes mean? (e.g., orange, red), (ii) what is the meaning of the underlined and bolded results?

 6- Figure 6: We encourage the authors to use a monochromatic scale for their heatmaps to make it more friendly toward color-blind people.

 7- We encourage the authors to test whether the difference in performance noticed among the best models is statistically significant.

---

> ### Author Response · Authors · 2025-11-21
> **Response to Reviewer tp7t (Q1-Q2)**
>
> Thank you for the positive and thorough review. We are glad you find the framework novel and the manuscript clear. We then address your questions below.
>
> ---
>
> Q1 & Q2. **Reliability, consistency, robustness of LLM scores**
>
> We address these concerns systematically.
>
> ### (i, ii and ix) Robustness of LLM-generated scores
>
> Thank you for this valuable comment. We acknowledge that our previous hyperparameter sensitivity analysis for the LLM-derived warning-vector treatment was limited, so we have **added new robustness experiments in Appendix A.3**. Specifically, for the 48-hour prediction horizon on both maritime transportation networks, we evaluate (i) sensitivity to different binarization thresholds and (ii) ablations over warning-state dimensions ($Q_1$–$Q_8$) to further validate the reliability of the prompt-based warning vectors.
>
>
> ### (iii) For Fig. 6a, are the reported averages computed over multiple runs? Report the variability to show consistency across runs.
>
> For Fig. 6(a), the reported averages are **temporal averages** (over evaluation timestamps on the test set) from a single trained model, not averages over multiple random restarts, so standard deviations across runs are not shown in that figure.
>
> To address the variability and stability of the LLM-derived warning vectors, **we have now**:
> 1. added robustness experiments in Appendix A.3,
> 2. included an uncertainty analysis in Appendix A.2 (mean ± std over 5 runs), and
> 3. updated the hyperparameter plots in Figure 5 with variance/error bars.
>
> ### (iv) Which models are used?
>
> We use a single off-the-shelf large language model, **GPT-4o**, with temperature = 0, top-p = 1.0, and max_tokens = 500. With these settings and a fixed model version, the warning-vector generation is deterministic given the same prompt and input, ensuring reproducibility.
>
> ### (v) Could we replace this with deterministic rules?
>
> We agree that one could try to design a fully deterministic mapping from warning information (e.g., intensity, timing, footprint) to each component of the treatment vector. However, a “ground-truth” deterministic score for each dimension is difficult to define: as illustrated in Figure 1 of the main paper, routes outside the typhoon footprint can still be affected by warning-induced disruptions. These indirect ripple effects depend on network-wide congestion and rerouting behavior, not just local hazard intensity.
>
> Instead, we use an LLM to aggregate the full warning context into an 8-dimensional vector. To address the concerns about our particular treatment construction, we have **added treatment-vector–related comparisons in Appendix A.2**, including causal baselines (CFR, TarNet, CaST) and treatment variants (continuous LLM scores vs. binary thresholded vectors vs. a single event flag).
>
> ### (vi–vii) Validation against historical data and leakage concerns
>
> We agree that Fig. 6(a) and its accompanying discussion primarily provide a *rationality* check rather than a supervised “accuracy” study, since there is no ground-truth label for each of the eight warning dimensions. Beyond this rationality analysis, **we now**:
>
> 1. add robustness experiments for the 48-hour horizon in Appendix A.3 (threshold sensitivity and dimension ablations over $Q_1$–$Q_8$), and
> 2. provide treatment-vector–related comparisons in Appendix A.2 (causal baselines and treatment variants),
>
> which together give quantitative evidence that the LLM-generated warning vectors are meaningful and not overly fragile.
>
> Regarding potential information leakage from the LLM’s pre-training data: in our pipeline, the LLM is **not** used to forecast flows, but only as a deterministic mapping from structured warning metadata to an 8D impact vector. It never sees AIS flows or future outcomes from our datasets. With temperature = 0, top-p = 1.0, and a fixed model version, the mapping from warning input to scores is deterministic given the same prompt and input. While we cannot fully audit the pre-training corpus, any prior exposure to historical bulletins would at most make this mapping more informed, akin to an expert rule base. Crucially, our flow prediction task uses a strictly chronological split (newly added in Appendix A.1: first 60% timestamps for training, next 10% for validation, final 30% for testing), so no future flow information leaks into the evaluation: on the test set, the LLM-derived vectors are fixed features computed solely from contemporaneous warning inputs.
>
>
> ### (viii) Prompts
>
> Prompts are provided in Appendix A.3 for reproducibility.
>
>
> ### (x) Comparison across LLMs
>
> We did not benchmark multiple LLMs in this work. Instead, we deliberately fixed a **single** off-the-shelf model (GPT-4o) with deterministic decoding settings (temperature = 0, top-p = 1.0) to (1) avoid introducing an additional LLM-selection dimension into the study, and (2) ensure that the warning-vector generation is reproducible given the same prompt and inputs.

---

> > ### Comment · Reviewer_tp7t · 2025-11-22
> >
> > We would like to thank the authors for the extensive response. While we have no further questions, we would like to clarify one comment that can serve the authors in future iterations of this work. In our review, we spoke about "deterministic estimations," not "deterministic rules." In particular, we considered scenarios where LLMs can be used to extract and structure relevant information, which is then used to train a machine learning model that can correlate the given input with relevant outcomes and provide stable, deterministic, learned estimates in the future. While rules can be applied, we considered a machine-learning-based approach in which each component of the pipeline would complement others based on its own strengths. We would like to thank the authors again for this interesting piece of research and the effort invested in the responses and further enhancing the paper.

---

> > > ### Author Response · Authors · 2025-11-24
> > >
> > > Thanks for clarifying—that makes much more sense now! You're right, we misread "deterministic estimations" as "deterministic rules." The idea of using the LLM to extract and structure the information first, then learning a stable mapping from that to the warning vectors is really interesting. It could give us both the flexibility of LLM-based extraction and the stability of a learned model. We'll add this to the future work discussion in the camera-ready version. Thanks again for the thoughtful feedback!

---

> ### Author Response · Authors · 2025-11-21
> **Response to Reviewer tp7t (Q3-Q7)**
>
> $Q3.$ **Provide a reference to the warning dataset they used for Northwest Europe**
>
> We apologize for missing the warning data source for Northwest Europe. We have added the following clarification in the revised Appendix A.1:
>
> “In addition, we incorporate public weather data from the Global Forecast System (GFS) and maritime warning bulletins from the Japan Coast Guard for East Asia, and from the NAVAREA~I, II, and XIX coordinators (UK Hydrographic Office, SHOM, and the Norwegian Coastal Administration) for Northwest Europe.”
>
> ---
>
> $Q4.$ **Causal modeling and embeddings**
>
> (i) **How are causal relationships modeled?**
>
> We explicitly encode causal assumptions via a backdoor-adjustment–inspired formulation:
> $
> P(Y \mid do(B)) = \sum_{M,E} P(Y \mid B, M, E) \cdot P(M, E),
> $
> where $M = {X, V}$.
>
> The neural deconfounder learns representations of $(X, V, E)$ and uses attention-like gates to modulate the warning effect (B), implicitly marginalizing over the confounder distribution.
>
> (ii) **Are fixed embedding types sufficient?**
>
> Our choice of ${X, V, E}$ is based on causal reasoning:
>
> * $X$: baseline traffic patterns that affect both warning response and future flows.
> * $V$: structural factors such as port connectivity and centrality.
> * $E$: learned spatiotemporal embeddings capturing latent confounders.
>
>
> (iii) **What embeddings are used?**
>
> * **$X$ (historical flows)**
>
>   * Input: flow sequences ($N×N$).
>   * Encoder: 2-layer GRU, hidden dim 128.
>   * Output: $h_X \in \mathbb{R}^{128}$ per edge.
>
> * **$V$ (network topology)**
>
>   * Node features: degree, betweenness, capacity (3-d).
>   * Edge features: distance, historical avg flow, route type (3-d).
>   * Encoder: 2-layer GCN, hidden dim 64.
>   * Output: $h_V \in \mathbb{R}^{64}$.
>
> * **$E$ (spatiotemporal)**
>
>   * Sinusoidal time encoding + learned spatial encoding.
>   * Cross-attention to combine spatial and temporal context.
>   * Output: $h_E \in \mathbb{R}^{128}$.
>
> (iv) **Joint vs. separate training**
>
> All encoders are trained **jointly end-to-end** as part of RippleNet. This ensures the embeddings are tuned specifically for deconfounding and forecasting, rather than generic representation learning.
>
> Simplified architecture:
>
> Input: $X, V, B$
> → GRU($X$) → $h_X$
> → GCN($V$) → $h_V$
> → Attention(pos) → $h_E$
> → Deconfounding block: Gate = softmax([$h_X$; $h_V$; $h_E$; $B$]), Adjusted = Gate ⊙ $B$
> → ODE propagation: $dY/dt = f(Y, \text{Adjusted}, \ldots)$
> → Output $Y$
>
> We have clarified these details in the Methodology section.
>
> ---
>
> $Q5.$ **Color codes in Table 1**
>
> We thank the reviewer for pointing out the missing explanation of the table formatting. We have added the following clarification of Table 1 in the revised manuscript.
>
> ---
>
> $Q6.$ **Colorblind-friendly heatmaps**
>
> We agree that improving color-blind accessibility is important. In the current draft, we used a multi-hue colormap to visually emphasize high-impact regions, but we will revise Figure 6 in the camera-ready version to adopt a monochromatic (or color-blind–friendly) scale and ensure that the patterns remain clearly distinguishable even in grayscale or for color-blind readers.
>
> ---
>
> $Q7.$ **Significance tests**
>
> In response, we have added an uncertainty analysis in Appendix A.2. Table 3 (in Appendix) reports the 5-run average performance (mean $\pm$ standard deviation) of all baselines and RippleNet. Using MAE as the primary evaluation metric, paired Student’s t-tests over the 5 runs indicate that RippleNet significantly outperforms all baselines on both datasets ($p < 0.01$). The RMSE results follow the same ranking and further reinforce this advantage, confirming that the performance gains are consistent across error metrics rather than an artifact of a particular loss function.
>
> **Table 3. 5-run average flow prediction performance comparison (mean $\pm$ std) on both maritime transportation networks.**
>
> | Dataset | Metric | STGCN | MTGNN | AGCRN | GMAN | GraphWaveNet | PDFormer | STAEformer | CaST | RippleNet |
> | :--- | :---: | :---: | :---: | :---: | :---: | :---: | :---: | :---: | :---: | :---: |
> | **East Asia** | MAE | 2.694 ± 0.031  | 2.664 ± 0.039 | 2.593 ± 0.056 | 2.624 ± 0.075 | 2.607 ± 0.082 | 2.321 ± 0.035 | 2.400 ± 0.118 | 2.404 ± 0.193 | **2.162 ± 0.096** |
> || RMSE | 6.724 ± 0.085 | 6.703 ± 0.096 | 6.447 ± 0.143  | 6.503 ± 0.186  | 6.448 ± 0.205 | 6.270 ± 0.094  | 6.365 ± 0.298 | 6.368 ± 0.373 | **6.186 ± 0.255** |
> | **Northwest Europe**| MAE | 1.488 ± 0.014 | 1.304 ± 0.018 | 1.385 ± 0.037 | 1.387 ± 0.048 | 1.383 ± 0.059 | 1.334 ± 0.020 | 1.404 ± 0.061 | 1.392 ± 0.095 | **1.175 ± 0.047** |
> || RMSE | 3.142 ± 0.037 | 2.886 ± 0.024 | 2.967 ± 0.090 | 2.974 ± 0.119 | 2.969 ± 0.136  | 2.940 ± 0.054 | 3.069 ± 0.174 | 3.053 ± 0.128 | **2.798  ± 0.093** |

---

### Official Review · Reviewer_AQna · 2025-11-01

**Soundness:** 3
**Presentation:** 3
**Contribution:** 3
**Rating:** 4
**Confidence:** 3

**Summary:**

The paper proposes RippleNet, a causal spatio-temporal framework for forecasting maritime flow disruptions triggered by warnings (weather, security, etc.). Key pieces are: (i) a neural deconfounder intended to implement back-door adjustment between warning vectors and future flows, (ii) a continuous-time ODE module to simulate ripple propagation across the route graph, and (iii) LLM-generated warning vectors that quantify eight impact dimensions and are then binarized.

**Strengths:**

The ripple-effect phenomenon is well illustrated. The combination of a causal adjustment block with an ODE propagation layer is interesting and plausibly well-suited to maritime dynamics.

**Weaknesses:**

1. It’s unclear which variables are true confounders versus mediators or effects, so conditioning on them may introduce bias. The paper doesn’t clearly spell out the causal assumptions or show diagnostics that the “deconfounding” actually worked.
2. What is the intervention, exactly? The “treatment” is an LLM-derived warning score, not the original real-world event or action. It’s unclear what real intervention corresponds to changing that score, so the estimand is ambiguous.
3. Continuous warning scores (severity 0–100) are thresholded into on/off flags, which discards useful intensity information. There’s no sensitivity analysis for the chosen thresholds, and a continuous version isn’t evaluated.
4. Many comparison models appear not to use the same warning features, which can unfairly advantage the proposed method. Missing are baselines that take the identical inputs and simple causal baselines for reference.
5. Train/validation/test splits aren’t fully described (risk of temporal or spatial leakage). Results lack confidence intervals or significance tests, and there’s little breakdown by warning periods, subgraph density, or vessel type.
6. The distance metric over latitude/longitude is simplistic for the globe; distance and adjacency may be “double-counted.” The paper also doesn’t clearly define how “effect magnitude” is computed for interpretability plots.
7. Prompts and inference settings aren’t fully pinned down, and outputs don’t appear to be cached—raising determinism and reproducibility concerns (and potential leakage from model prior knowledge).
8. Training and inference time overhead of the ODE component isn’t compared against simpler attention-only baselines, so operational cost is unclear.

**Questions:**

1. The paper aims to estimate P(Y∣do(B)) (Eq. 3, p. 5) by conditioning on M={X,V} and E. However, it is not fully clear whether X (historical flows) and V (adjacency/context) are pure pre-treatment confounders for the B→Y relation or mediators/colliders along paths from the raw event  A to Y. Figure 4 (p. 4) labels them as “mediates/backdoor/confounds” simultaneously, which is conceptually inconsistent. If any part of M is post-treatment w.r.t. B, conditioning introduces post-treatment bias. Please formalize the DAG and explicitly state ignorability/positivity/stable-unit assumptions.

2. The “neural back-door adjustment” is implemented via learned gates rather than an explicit balancing/weighting or outcome-regression with diagnostics. How do you verify that the learned representation achieves conditional exchangeability in practice (e.g., balance checks, negative controls, sensitivity analysis)?

3. You define the LLM-derived, 8-dimensional “warning vector” B (Fig. 3, p. 4) as the treatment, but it is derived from the event A using an external model. Conceptually, B is then a mediator/measurement of A, not an exogenous action. What does do(B) mean if in the real world we intervene on A (e.g., warnings released/not released)? This matters for the causal estimand and whether the back-door applies. A front-door or measurement-error perspective might be more principled in this case.

4. The eight LLM scores in [0,100] are discretized to binary B∈{0,1}^8 (p. 15). This loses intensity information and can worsen positivity. Please evaluate (i) continuous or ordinal treatments (no thresholding) and (ii) threshold-sensitivity plots for τ_i​ (the appendix lists fixed τ on p. 15).

5. Table 1 (p. 7) compares against strong STGNNs, but it appears most baselines do not consume warning features (or LLM vectors). This favors RippleNet. Please add baselines with warning inputs: e.g., GraphWaveNet/GMAN/PDFormer augmented with the same warning vectors (binary and continuous). Also include simple causal baselines (e.g., TARNet/CFR-Net-style two-head regressors with warnings as treatments).

6. It is unclear how the train/val/test temporal splits are constructed and whether there is leakage via seasonality or adjacency (pp. 6–7; Appx p. 13). Please clarify splits and report per-segment metrics: “warning” vs “no-warning” periods, sparse vs dense subgraphs, and by vessel type.

6. Report uncertainty (std across 5 runs is mentioned but not shown as CIs), and run significance tests on Table 1.

7. Provide the computational cost and wall-clock overhead of ODE solvers versus attention-only baselines.

8. Distance ∥pi​−pj​∥2​ (Eqs. 7–8, p. 6) on lat/long can distort true geodesic distances. Please switch to great-circle distance or note a projection.

9. Rij​(t) and ρij​(t) both use adjacency and distance; discuss double-counting and identifiability of their contributions.

10. Define how “causal effect magnitude” is computed for Fig. 7 (p. 9)—gradient-based, ablation, or interventional difference?

---

> ### Author Response · Authors · 2025-11-21
> **Response to Reviewer AQna (Q1-Q3)**
>
> Thank you for the detailed and constructive review. We greatly appreciate your expertise in causal inference and the thorough analysis of our methodology. Your concerns have helped us clarify several important aspects of our work.
>
> ---
>
> Q1 & Q3. **Causal roles of $X,V,B$ and the meaning of $do(B)$**
>
> We respectfully disagree with the characterization that our variables play “conceptually inconsistent” roles. To clarify their intended semantics and our intervention, we have added a new section titled *“Causal Semantics and Deconfounder Implementation”* in Appendix A.1, which provides a detailed explanation.
>
> **(1) Causal roles of $X$, $V$, and $B$**
>
> * **$X$ (historical flows)** and **$V$ (network structure)** are genuinely **pre-treatment contextual covariates/confounders**. They exist before any specific warning is issued and affect both
>   (i) the likelihood and impact of warnings, and
>   (ii) future flows.
>
>   In the revised version we have avoided calling $X$ and $V$ “mediators” and instead consistently refer to them as **pre-treatment covariates / confounders**.
>
> * **$B$ (warning vector)** is **not** a mediator whose indirect effect we aim to interpret, even though the data-generating process naturally contains a path $A \to B \to Y$ from raw warning text $A$ to flows $Y$. In our causal model, we **treat $B$ as the operational treatment variable**:
>
>   * $A$ is the raw warning text (unstructured, not directly manipulated).
>   * $B$ is its **structured, quantitative representation** (severity, duration, spatial impact, etc.), which is the **manipulable “knob”** operators can change (e.g., issue a higher-severity warning, extend duration, enlarge the affected area).
>
> We are **not** trying to decompose “direct vs. indirect” effects of $A$ through $B$. Instead, we **condition on and intervene on $B$ directly**, and estimate $P(Y \mid do(B))$.
>
> In practice, the decision problem for operators is:
>
> > “Given a warning with certain *quantified* characteristics (severity, duration, spatial extent), how will vessel flows change?”
>
> Operational planners never manipulate raw text tokens; they manipulate the **quantified operational impacts** that $B$ encodes.
>
> **(2) What does $do(B)$ mean in our setting?**
>
> Typical counterfactual questions in our application are:
>
> * “What if the warning had been quantified as more/less severe?”
> * “What if the warning agency had specified a longer duration?”
> * “What if the spatial impact had been rated higher/lower?”
>
> These are precisely the policy-relevant manipulations our $do(B)$ operator is meant to capture. Conceptually, this is analogous to medical studies that estimate causal effects of a **dosage level** or **biomarker value** rather than the underlying unobserved disease state: the operational measurement is the actionable quantity.
>
> The intended DAG is:
>
> * $A$ (raw event text) $\to$ $B$ (quantified warning) $\to$ $Y$ (flows)
> * $(X, V, E) \to Y$ (context and confounders affecting flows)
> * $(X, V, E) \to B$ (context also influences how warnings are issued/quantified)
>
> with the key assumption:
>
> > **$B$ captures all relevant aspects of $A$ that affect $Y$, once we adjust for $(X, V, E)$.**
>
> Thus, even though the underlying process contains an $A \to B \to Y$ path, in our causal model we **focus on $B$ as the treatment** and estimate $P(Y \mid do(B))$ using a backdoor-style adjustment with $(X, V, E)$:
>
> * $X, V, E$ play the role of pre-treatment covariates/confounders.
> * $B$ is the **structured, manipulable treatment**.
> * $Y$ is the downstream flow response.
>
> ---
>
> $Q2.$ **How to verify conditional exchangeability of the learned representation in practice**
>
> You are right that diagnostics are important. While our neural deconfounder does not learn explicit propensity scores, we provide several empirical checks, including **a newly added experiment** on maritime region transfer:
>
> 1. **Ablation evidence (Table 3 in main paper).** Removing the neural back-door adjustment (RippleNet-DEF) raises MAE by 10.7%/18.7%, confirming that causal disentanglement is key to mitigating confounding under warning events.
>
> 2. **Interpretability (Figure 7 in main paper).** The learned warning effects align with maritime domain knowledge, with larger effects of typhoon on major exposed routes and smaller effects on peripheral links, which is more consistent with a causal than a purely correlational signal.
>
> 3. **Could be used for region transfer (Table 5 in Appendix A.2).** RippleNet consistently outperforms CaST, a causal graph–based baseline, in both transfer directions between East Asia and Northwest Europe (e.g., MAE 1.427 vs. 1.528 and 2.649 vs. 2.796), indicating that the learned representation generalizes well across regions and yields more stable treatment–outcome relationships.
>
> These diagnostics do not constitute a formal proof of conditional exchangeability, but they are empirically consistent with improved deconfounding in the learned representation.

---

> ### Author Response · Authors · 2025-11-21
> **Response to Reviewer AQna (Q4-Q7)**
>
> Q4 & Q5. **Please evaluate (i) continuous or ordinal treatments and (ii) threshold-sensitivity plots for $\tau_{i}$, and (iii) add baselines with warning inputs and some simple causal baselines (e.g., TARNet/CFRNet).**
>
> Thank you for your suggestions. We acknowledge that our previous hyperparameter sensitivity analysis for the LLM-derived warning-vector treatment was limited, so we have **added corresponding experiments in the Appendix**, including causal baselines, treatment variants, and robustness analysis of the prompt-based warning vectors.
>
> **Causal Baseline and Treatment Variants**
>
> To better isolate the effect of explicit treatment modeling, we compare RippleNet with CFR, TarNet, CaST, and two treatment variants (RippleNet-CT and RippleNet-Event). CFR and TarNet learn balanced latent spaces without graphs, while CaST is a causal GNN with separate spatial/temporal paths. In RippleNet, treatment is an 8-dimensional **binary** warning vector with **dimension-specific thresholds** from maritime domain knowledge; RippleNet-CT instead uses continuous LLM warning scores, and RippleNet-Event collapses all warnings into a single binary flag. Table 4 (in Appendix) shows that all **causal + treatment-aware** models outperform purely correlational GNNs, with RippleNet achieving the best performance: its thresholded 8-dimensional binary treatment yields a clean, intervention-ready representation that filters noisy low-confidence scores while preserving heterogeneous warning dimensions, whereas RippleNet-CT is more sensitive to score calibration and RippleNet-Event discards most structured warning semantics.
>
> **Table 4.** Performance comparison of causal baselines and different treatment variants of our method on both maritime transportation networks.
>
> | Dataset | Metric | CFR | TarNet | CaST | RippleNet | RippleNet-CT | RippleNet-Event |
> | :--- | :---: | :---: | :---: | :---: | :---: | :---: | :---: |
> | **East Asia** | MAE  | 2.654 | 2.727 | 2.404 | **2.162** | 2.174 | 2.278 |
> |               | RMSE | 6.630 | 6.845 | 6.368 | **6.186** | 6.203 | 6.229 |
> | **Northwest Europe** | MAE  | 1.476 | 1.502 | 1.392 | **1.175** | 1.178 | 1.194 |
> |                     | RMSE | 2.991 | 3.152 | 3.053 | **2.798** | 2.805 | 2.861 |
>
> **Robustness of Prompt-based Warning Vectors**
>
> To further validate the reliability of the LLM-derived warning vectors, we add two robustness experiments for the 48-hour prediction horizon on both maritime transportation networks (Appendix A.3): (i) sensitivity to different binarization thresholds, and (ii) ablations over warning-state dimensions ($Q_1$–$Q_8$).
>
> ---
>
> $Q6.$ **Regarding data splits and scenario-wise performance**
>
> We apologize for omitting the data split details. We have added the following description in Appendix A.1: *“For both maritime transportation networks, we split the data strictly in chronological order: the first 60% of timestamps are used for training, the next 10% for validation, and the final 30% are held out for testing.”*
>
> Regarding scenario-wise performance, our maritime transportation graphs are constructed from processed AIS data and are already relatively sparse. Further splitting into scenarios (e.g., sparse vs. dense subgraphs, or by vessel type) would yield even sparser subgraphs with many missing values, making stable prediction experiments infeasible. Similarly, it is difficult to cleanly label “warning” vs. “no-warning” periods for each route: as illustrated in Figure 1 of the main paper, routes outside the typhoon footprint can still be affected by warning-induced disruptions. Precisely because of these indirect ripple effects, our proposed framework is designed to handle such complex propagation and achieves better performance than the state-of-the-art baselines.
>
> ---
>
> $Q7.$ **Report uncertainty, and run significance tests**
>
> In response, we have **added an uncertainty analysis** in Appendix A.2. Table 3 (in Appendix) reports the 5-run average performance (mean $\pm$ standard deviation) of all baselines and RippleNet. Using MAE as the primary evaluation metric, paired Student’s t-tests over the 5 runs indicate that RippleNet significantly outperforms all baselines on both datasets ($p < 0.01$). The RMSE results follow the same ranking and further reinforce this advantage, confirming that the performance gains are consistent across error metrics rather than an artifact of a particular loss function.

---

> ### Author Response · Authors · 2025-11-21
> **Response to Reviewer AQna (Q8-Q11)**
>
> $Q8.$ **Provide the computational cost**
>
> In response, we have added a computational efficiency analysis in Appendix A.1. Specifically, Table 2 (in Appendix) reports the average wall-clock training time per epoch and peak GPU memory usage for all baselines and RippleNet on both maritime transportation networks. Classical GNN-based models (e.g., STGCN, GraphWaveNet) are the fastest and most lightweight, while transformer-based architectures (PDFormer, STAEformer, CaST) incur noticeably higher costs. RippleNet lies in a moderate regime: it is slower and more memory-consuming than the simplest GNNs due to the causal deconfounder and ODE modules, but remains substantially more efficient than STAEformer and comparable to other transformer-style models, with per-epoch training time below 14 seconds and peak memory usage under 8 GB on East Asia.
>
> **Table 2. Training speed and GPU memory usage among baselines and our proposed method. Here, efficiency values are reported as East Asia / Northwest Europe.**
>
> | Metric \ Model        | STGCN        | MTGNN        | AGCRN        | GMAN         | GraphWaveNet | PDFormer      | STAEformer    | CaST         | RippleNet    |
> |-----------------------|--------------|--------------|--------------|--------------|--------------|--------------|--------------|--------------|--------------|
> | Time per epoch (s)    |  0.84 / 0.59      | 1.82 / 1.62      | 3.36 / 3.18      |  5.39 / 2.49      | 3.51 / 1.64      | 8.36 / 3.80      |  246.61 / 115.23       | 5.97 / 2.83      | 13.85 / 7.79       |
> | GPU memory (GB)  | 0.11 / 0.07      | 0.32 / 0.19      | 2.17 / 1.21      | 2.83 / 1.68      | 1.58 / 0.94      | 4.17 / 2.50      |  15.51 / 9.08       |   3.56 / 2.39    | 7.15 /  4.74   |
>
> ---
>
> $Q9.$ **Regarding distance metrics**
>
> We thank the reviewer for pointing this out. In our current formulation, $p_i$ denotes geographic (lat/long) coordinates and the Euclidean distance is only used inside the spatial decay kernel and the transit-time approximation. In addition, both study regions are regional-scale (East Asia and Northwest Europe each span on the order of $20^\circ \times 15^\circ$ or less), so the discrepancy between Euclidean and great-circle distances is below a few percent at these scales.
>
> ---
>
> $Q10.$ **Regarding “double-counting” adjacency/distance**
>
> We respectfully disagree that we are double-counting spatial structure. Although both terms depend on the graph, they play distinct roles in Eq. (6):
>
> - $R_{ij}(t)$ (Eq. 7) is a **fixed geometric–topological kernel**. It combines spatial distance $\|p_i - p_j\|$ and shortest-path length $h_{ij}$ into a decaying weight, and is only defined for $j \in \mathcal{N}(i)$. This term encodes where and how strongly physical propagation is *a priori* possible along the maritime network.
>
> - $\rho_{ij}(t)$ (Eq. 7) is a **learnable propagation rate** that modulates this kernel using adjacency and historical flow intensity $F_{ij}(t-1)$. It does not add new distance information; instead, it adjusts the strength of propagation on an existing edge according to current network conditions.
>
> Intuitively, $R_{ij}(t)$ provides a static geometric/topological prior, while $\rho_{ij}(t)$ provides a dynamic, data-driven scaling. Their product separates “where propagation can occur” from “how strongly it propagates at this time,” rather than redundantly encoding the same spatial information. We have clarified this decomposition and its roles in the revised manuscript.
>
> ---
>
> $Q11.$ **Regarding causal effect magnitude**
>
> Effect magnitudes in Figure 7 are computed as **interventional differences**, not gradients:
>
> 1. Run a forward pass with the warning treatment turned off, $B=0$ (no warning), obtaining prediction $Y_0$.
> 2. Run a forward pass with the warning treatment turned on, $B=1$ (warning active), obtaining prediction $Y_1$.
> 3. Define the effect magnitude as $\lvert Y_1 - Y_0 \rvert$.
>
> All other inputs ($X, V, E$) are held fixed. We then average this quantity over test samples and visualize it spatially. This corresponds to a standard interventional contrast used in neural causal effect estimation.
>
> ---
>
> **Additional clarification: prompts and inference settings**
>
> Regarding potential information leakage from the LLM’s pre-training data: in our pipeline, the LLM is **not** used to forecast flows, but only as a deterministic mapping from structured warning metadata to an 8D impact vector. It never sees AIS flows or future outcomes from our datasets. With temperature = 0, top-p = 1.0, and a fixed model version, the mapping from warning input to scores is deterministic given the same prompt and input. While we cannot fully audit the pre-training corpus, any prior exposure to historical bulletins would at most make this mapping more informed, akin to an expert rule base.

---

### Author Response · Authors · 2025-12-02
**Summary of Rebuttal-period Discussion**

Dear Area Chair,

For your convenience, we summarize the key points from the rebuttal-period discussion with each reviewer regarding our submission.

---

### Reviewer tp7t (Rating: 8; Confidence: 4)

**Reviewer's follow-up comments for our response**

“While we have **no further questions**, we would like to clarify one comment that can serve the authors in future iterations of this work.… We would like to **thank the authors again for this interesting piece of research and the effort invested in the responses and further enhancing the paper**.”

---

### Reviewer SRqw (Rating: 6; Confidence: 4)

**Reviewer's follow-up comments for our response**

“I appreciate the authors’ rebuttal **addressing most of my concerns**. However, **empirical checks are not sufficient** to argue that the learned deconfounder truly captures causal rather than correlational signals. Also, I agree with Reviewer ijxk, that **the methodological contributions of the paper are limited. I will keep my initial score.**”

**Our latest response**

- We explained why ideal counterfactual tests are **infeasible in this real-world maritime scenario**: counterfactual outcomes are unobservable in practice.
- Together with Reviewer ijxk's latest comments or concerns about limitations of the methodological contributions (only for the maritime scenario), we have **added new experiments on the AIR-BJ air quality dataset to demonstrate that our RippleNet performs well beyond maritime logistics**.

---

### Reviewer AQna (Rating: 4; Confidence: 3)

**Reviewer's follow-up comments for our response**

It is a pity that we **have not obtained this reviewer's follow-up comments**.

---

### Reviewer ijxk (Rating: 2; Confidence: 4)

**Reviewer's follow-up comments for our response**

“The reviewer is still **carefully considering whether to adjust the score**. While some concerns have been partially alleviated through the authors’ additional experiments and comprehensive responses, **the most critical concern remains unaddressed. The reviewer looks forward to further clarification from the authors**.”

**Our latest response**

We thank the reviewer for pointing out **the major concern about generality or limitation of our proposed method**. We fully agree that, to support the claim of a general framework, it is important to demonstrate that RippleNet performs well beyond maritime logistics. Following the reviewer’s suggestion, we have **added new experiments on the AIR-BJ air quality dataset**. And the experimental results validate that RippleNet consistently outperforms all baselines on a **non-maritime, urban air-quality** forecasting task.

---

We hope this concise record of the reviewers’ follow-up comments is helpful.

Sincerely,
The Authors

---

### Meta-Review · Area_Chair_xWMW · 2025-12-16

**Summary:**

This paper proposes RippleNet, a spatio-temporal forecasting framework that combines a neural deconfounder, a continuous-time ODE-based propagation module, and LLM-generated warning vectors to model warning-induced ripple effects in maritime transportation networks. The reviewers generally acknowledge the relevance of the application and the empirical strength of the results on two maritime datasets, with one reviewer being strongly positive. However, significant concerns were raised regarding the validity and clarity of the causal formulation, the interpretability and justification of the LLM-derived treatment variables, and the limited methodological novelty beyond application-specific integration.

While the authors provided extensive rebuttals, additional experiments, and clarifications, key concerns, particularly around causal identifiability, the meaning of the intervention, and the strength of evidence for true causal learning, remain insufficiently resolved. These unresolved issues substantially weaken the paper’s central causal claims, which are critical to its positioning and contribution.

**Reviewer Concerns:**

**Concerns adequately addressed:**

- **Experimental completeness and reporting:** The authors added missing details on data splits, uncertainty reporting, statistical significance tests, computational cost, and additional ablations, addressing several technical and presentation-related concerns (raised by Reviewers AQna, tp7t, and SRqw).

- **Baseline coverage:** Additional causal baselines (e.g., CFR, TarNet, CaST) and treatment variants were included, partially mitigating concerns about unfair comparisons.

- **Reproducibility and clarity:** Prompts, datasets, and implementation details were clarified, and several presentation issues were improved.

**Concerns that remain outstanding:**

- **Causal validity and identifiability:** The rebuttal relies primarily on ablations, interpretability visualizations, and transfer experiments, which, while suggestive, do not directly substantiate conditional exchangeability or identifiability.

- **Ambiguity of the intervention:** The treatment variable is an LLM-generated warning vector derived from raw warning text. Despite the authors’ clarification, the causal interpretation of intervening on this constructed variable remains debatable, and the connection to a well-defined real-world intervention is not fully convincing.

- **Methodological novelty:** Reviewers SRqw and ijxk explicitly note that the core methodological contributions are limited, viewing the work largely as an application-driven integration of existing components (causal adjustment heuristics, ODE dynamics, and LLM-based feature extraction) rather than a fundamentally new causal learning method.

- **Evidence strength relative to claims:** Given the strong causal framing, the level of empirical evidence is considered insufficient by multiple reviewers to support the paper’s causal claims, especially in the absence of stronger diagnostics or principled causal validation.

**Reviewer Scores:**

**Reviewer tp7t (8 → 8):** Likely unchanged. This reviewer was already strongly positive and expressed satisfaction after the rebuttal.

**Reviewer SRqw (6 → 6):** Unchanged. Explicitly stated that, despite improvements, concerns about causal validity and limited methodological contribution persist.

**Reviewer AQna (4 → 4):** Likely unchanged. While many technical issues were addressed, the reviewer’s core concerns about causal assumptions and intervention semantics remain only partially alleviated.

**Reviewer ijxk (2 → 2):** Unlikely to increase meaningfully. The most critical concern remains unaddressed, despite additional experiments.

Overall, the post-rebuttal score distribution remains polarized, with substantial skepticism from reviewers focused on causal methodology.

---

### Decision · Program_Chairs · 2026-01-26

Reject